# Volume Uptake of Carbonyls during Diffusional Ice Crystal Growth

Jackson Seymore*[1], Miklós Szakáll[1], Alexander Theis[3], Subir K. Mitra[3], Christine Borchers[2], Thorsten Hoffmann[2]

[1] Institute for Atmospheric Physics, Johannes Gutenberg University, Mainz, Germany
[2] Department of Chemistry, Johannes Gutenberg–University, Mainz, Germany
[3] Particle Chemistry Department, Max Planck Institute for Chemistry, Mainz, Germany

*Corresponding author: Jackson Seymore seymorej@uni-mainz.de

**Keywords**: volume uptake, gas-ice partitioning, diffusional crystal growth, Orbitrap MS, secondary organic aerosol, SOA, convective clouds, entropy-enthalpy compensation, EEC

## Abstract

Carbonyls are highly relevant atmospheric constituents that influence tropospheric photochemistry and oxidative capacity. They can be removed from the upper troposphere via ice phase deposition scavenging. The gas-volume uptake coefficients for 14 different carbonyl compounds were determined using a flowtube apparatus. Ice crystals were grown from vapor deposition in the presence of gas phase carbonyls at –20, –30, and –40 ºC. Using van't Hoff analysis, the entropy and enthalpy of uptake were determined. An inverse relationship between uptake coefficients and temperature was observed for all species except methyl vinyl ketone. A linear correlation between $\Delta S$ and $\Delta H$ arose which was statistically validated and determined with 99% confidence to not be a statistical artifact. This compensation behavior could be an indication of a surface liquid layer or quasi-liquid layer behavior involved in the uptake process and could also indicate a single dominant influence on a compound's uptake. The most significant physicochemical properties correlated with uptake were identified to be vapor pressure and molar mass, which indicate that smaller compounds with higher vapor pressures are more readily taken into the ice phase. The volume uptake coefficients observed here are below the 10 mol m$^{-3}$ Pa$^{-1}$ threshold given by Crutzen and Lawrence (2000) to be considered a substantial atmospheric removal process.

## 1 Introduction

Scavenging of organic gases by hydrometeors—such as rain, snow, graupel, and cloud droplets—has a cleaning effect on the atmosphere. While Henry's law coefficients are available to describe the interactions with

most organic gases with liquid water (Sander, 2023), less is known about its interactions with ice phase deposition. Most precipitation in the midlatitudes as well as all cloud formation in the upper troposphere is formed via ice and subsequently indicates that ice is a significant contributor to the wet deposition of trace atmospheric constituents (Franz and Eisenreich, 2000; Heymsfield et al., 2020; Mülmenstädt et al., 2015). Ice growth in the atmosphere operates under a few distinct processes: (1) the collection of supercooled cloud droplets by ice (riming), (2) direct

liquid freezing, and (3) vapor-to-ice growth by diffusion. The latter process—often referred to as depositional growth—is hereafter referred to as diffusional growth and is the responsible mechanism for all cloud formations in the upper troposphere (Heymsfield et al., 2020; Mülmenstädt et al., 2015). There has been significant research in recent years on the redistribution and revolatization of organics during riming or liquid freezing (Borchers et al., 2024; Jost et al., 2017; Gautam et al., 2025; Seymore et al., 2025), but very little for diffusional growth.

Carbonyls as a class of trace atmospheric constituents are highly relevant secondary organic aerosol (SOA) precursors and intermediates (Ervens and Kreidenweis, 2007; Galeazzo et al., 2024; Srivastava et al., 2022; Yu et al., 2014). They are ubiquitous and play vital roles in tropospheric photochemistry and oxidative capacity, which affects radical cycling and ozone formation (Xu et al., 2023). Specifically, the photolysis of carbonyls is an important source of peroxy radicals in the atmosphere. After photolysis, the aldehydic group (-CHO) decomposes and forms

the $HO_2$ radical with the addition of $O_2$. This peroxy radical is then a source of atomic oxygen for ozone formation (Liu et al., 2022). Formaldehyde, acetaldehyde, and acetone are typically the most abundant and considered the main contributors to ·OH reactivity and dominate the ozone formation potential of the total oxygenated volatile organic compounds (VOC), while less abundant species like benzaldehyde make a minor contribution to the ·OH removal rate and inhibit ozone formation (Wang et al., 2022). Additionally, lower vapor pressure carbonyls can be oxidized

to form SOA through gas-particle partitioning. This chemical evolution and physical transformation results in a range of lifetimes for various carbonyls from hours (e.g., below 1 hr for unsaturated aldehydes against OH oxidation, 1–3 hr lifetime for glyoxal against photolysis and OH oxidation, 1.3 hr for formaldehyde and glyoxal under overhead sun conditions) to a few days (i.e., 15 days for acetone with respect to the oxidation by OH) depending on their structures (Jacob, 2021). Compounds like glyoxal and methylglyoxal are significant contributors

to SOA formation via aqueous-phase chemistry (Ling et al., 2020) and many other such SOA contributing carbonyls are isoprene products, like methacrolein, methyl vinyl ketone (MVK), hydroxyacetaldehyde, or hydroxyacetone (Grosjean et al., 1993). Nopinone, diacetyl, camphor, norcamphor, and propionaldehyde are other naturally

occurring carbonyls of interest that present a variety of structures and properties for comparison. Notably nopinone, camphor, norcamphor, and propionaldehyde are analogous bicyclic monoterpenoids while diacetyl is another

diketone similar to glyoxal and methylglyoxal.

Despite their relevance, there are limited studies describing their removal from the upper atmosphere via deposition scavenging and only then describe their resultant wet deposition (Bartels-Rausch et al., 2014; Mu and Xu, 2009). Huffman and Snider (2004) attempted to measure uptake of acetone as a representative for ketones as hydrogen bond acceptors, however background contamination prevented thorough characterization. They did,

however, publish a volume uptake coefficient for acetone and concluded that it had a significantly lower uptake than the alkanols studied.

The publications by Fries et al. (2007) and Huffman and Snider (2004) concern the ice uptake of aromatic hydrocarbons and oxyhydrocarbons. These are the first studies to describe the interactions between diffusion-growing ice and depositing organic vapors. Outside of these, the only measurements for diffusional uptake are for

select inorganic species (e.g. $H_2O_2$, HCl, $HNO_3$, etc.) (Bartels-Rausch et al., 2014; Conklin et al., 1993; Diehl et al., 1995; Dominé and Thibert, 1996; Mitra et al., 1990; Santachiara et al., 1998) or adsorption on nongrowing ice (Abbatt et al., 2008; Von Hessberg et al., 2008). These experiments to evaluate gas interactions with ice describe either "growing ice" or "static ice" which have investigated the effects of volume uptake and surface processes respectively. From this distinction, these publications reveal that many uptake processes are predominately volume

uptake with secondary contributions from certain surface processes, notably (1) bonding to the air-ice interface or (2) uptake into a liquid solution phase coexisting with ice (Conklin et al., 1993; Goss, 1993). Further, they show that equilibrium treatments of both surface and volume uptake can correctly predict the resulting concentrations in snow (Dominé and Thibert, 1996).

Treating the uptake of organic compounds by ice crystals growing through diffusion as an equilibrium

allows for the calculation of gas-ice partitioning coefficients, referred hereafter as uptake coefficients. These coefficients ($K_{g,ss}$) describe the ratio of the gas phase concentration of the analyte vapor to its concentration in the ice phase. More specifically, with the possibility for a liquid solution phase that coexists with ice, this relationship is described as:

$K_{g,ss} = K_{g,l}K_{l,ss}$ (1)

where the equilibrium constants $K_{l,ss}$ and $K_{g,l}$ relate the analyte concentrations in the ice solid solution (*ss*) and liquid

water (*l*) phases to the analyte partial pressure in the gas phase (*g*). The last equilibrium constant ($K_{l,ss}$) relates the

concentrations in the liquid water (*l*) and solid solution (*ss*) phases. The present study reports direct measurements of

$K_{g,ss}$ for carbonyl compounds and neglects thorough investigation of gas to liquid or liquid to ice equilibrium.

Hereafter, $K_{g,ss}$ will be referred to simply as *K* and is specifically considered a volume uptake coefficient.

In the present study, laboratory experiments explore the uptake of 14 different carbonyl species by ice

crystals during vapor deposition growth. Ice crystals were grown from vapor deposition under controlled humidity

conditions in the presence of gaseous carbonyl species. The blended gas mixture was targeted to produce roughly 10

ppbv of each gaseous analyte to maintain analytical reliability, approximately 2–3 orders of magnitude larger than

the partial pressures of these compounds in the unpolluted troposphere. Water vapor saturation was controlled to

50% supersaturation (wrt ice) to achieve realistic growth conditions in natural cirrus clouds. Gas phase

concentrations were determined using an integrative denuder technique and subsequent derivatization to aid

detection for both gas and ice phase concentrations. Ultra-high performance liquid chromatography with ultra-high

resolution mass spectrometry (UHPLC-UHRMS) was then used to analyze the samples and determine their uptake

coefficients.

**2 Methods**

**2.1 Experimental Setup**

    The experimental design in this paper is a variation on the experiment presented by Fries et al. (2007) with

gas measurement techniques developed by Kahnt et al. (2011). In the present experiment, ice crystals are grown in

the presence of carbonyl vapors. Their concentrations in the ice-phase and the gas phase are then measured to

determine their uptake coefficients.

    Three uptake experiments were performed with three replicates at atmospheric pressure with the

experimental apparatus shown in Figure 1. The setup had three main stages: an ambient temperature gas-mixing

stage, a chilled crystal growth flowtube, and outflowing gas measurement. In the first stage (Stage 1. Gas Mixing),

pressurized dry nitrogen gas was passed through a bubbler with known concentrations of the analytes in aqueous

solution. This stream of gas then reached saturation at ambient temperature (maintained at 23 ⁰C and confirmed by

measurment) and passed through a glass frit and a droplet catching chamber to ensure no liquid droplets remained in

the gas stream. The saturated gas stream was then diluted with dry nitrogen to reach the desired humidity and vapor

concentration. The specific saturation and vapor concentration mix was maintained using a Mass Flow Controller

(Brooks Instrument B.V. 5850TR/FA1B201). The resulting mixed gas was allowed to homogenize inside a 0.5 L

mixing chamber and then was introduced into the insulated chamber with the crystal growth tube. Another Mass

Flow Controller was used to ensure a constant volumetric flowrate was maintained through the flowtube and

through a branching line for input gas measurement.

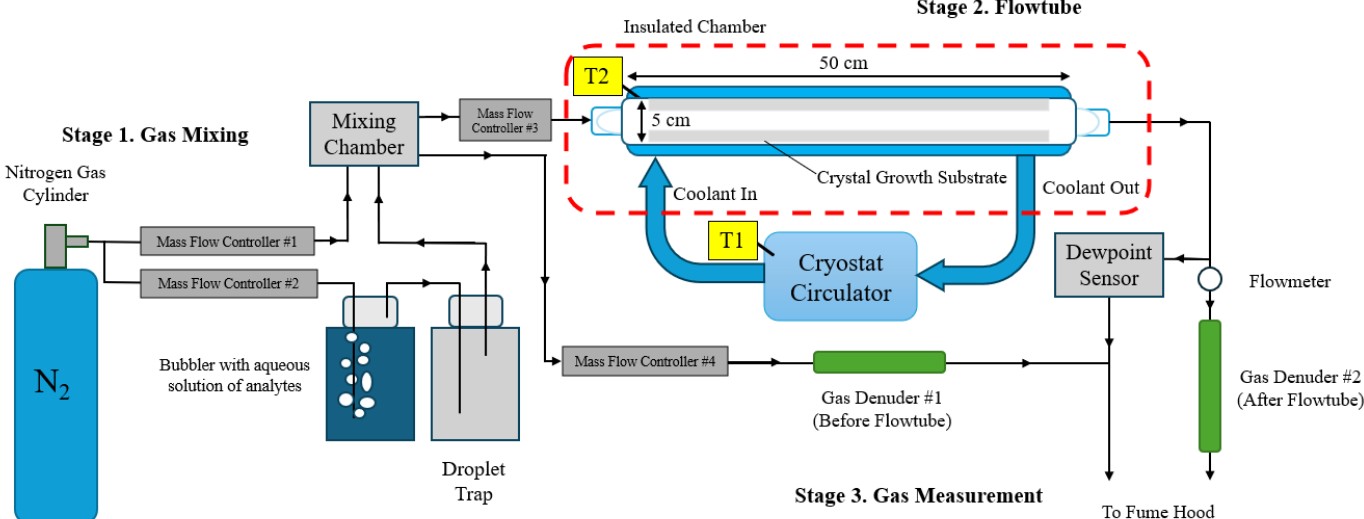


**Figure 1. Experimental Apparatus Design**

        The second stage of the apparatus is the flowtube portion of the setup (Stage 2. Flowtube). The flowtube is

a quartz-glass tube of 50 cm length and 5 cm diameter with an approximate interior volume of 1 L where ice crystals

were grown from diffusional vapor deposition. It is capped at both ends with a uniform flow PTFE nozzle with

double silicon VMQ O-rings to ensure proper sealing of the gas stream at low temperatures. The flowtube is inserted

into a copper thermal exchange coil with a PT100 temperature sensor (T2) mounted between the coil and the tube.

The flowtube and coil are housed inside an insulated chamber while ethanol coolant is circulated through the coil

from a cryostat circulator (Julabo F81) outside the chamber. Using the coolant bath temperature (T1) and T2, the

temperature of the flowtube glass substrate can be maintained within 0.1 C of the target temperature using PID

control. Prior to each experiment, the flowtube substrate is ethanol cleaned, dried with nitrogen gas, and brought to

the target temperature under the flow of dry nitrogen. This prevents ice nucleating contaminants and ambient

humidity from depositing on the substrate prior to exposure to the test gas. The flowtube is allowed to cool for 1

hour prior to exposure to ensure temperature stability. This setup was also thermally characterized using a PT100

sensor mounted on a probe inserted along the center of the tube. The quantitated temperature error was determined

using this method and the resulting temperature profiles can be found in Figure S1 of the Supplemental Materials.

After the flowtube is the third stage of the setup where the effluent gas is analyzed (Stage 3. Gas

Measurement). Continuous dewpoint measurements along with an integrative measurement of vapor analytes were

taken on the before and after flowtube gas streams using a dewpoint hydrometer (Michell Instruments S8000) and a

pair of reagent-coated gas denuders. Ice samples were collected from the flowtube, weighed, and analyzed for their

analyte concentration. This data is then able to be used to calculate the uptake coefficient. PTFE tubing was used for

all the tubing connections and the flow conditions in these experiments remained laminar with a Reynolds number

of approximately 160, which is well below the critical value of 2300 for pipe flow (Warhaft, 1998). Experiments

were performed 24 hours at –20 and –30 ºC, 48 hours for –40 ºC to allow for adequate crystal growth and saturation

of any vapor wall losses.


**2.2 Crystal Growth Conditions**

Ice crystals were grown from vapor deposition under controlled humidity conditions in the presence of

gaseous carbonyl species. For the experimental temperatures of –20, –30, and –40 ºC, ice was grown at 50%

supersaturation (wrt ice) with 10 ppbv of analyte vapor (140 ppbv total organics). To achieve these target conditions,

different dilutions of dry nitrogen with the saturated gas stream were mixed using different bubbler solutions with

the required aqueous concentrations of analytes to reach the target partial pressure. The saturation pressure of water

vapor over ice ($e_i$) was calculated using the Sonntag parameterization (Sonntag, 1994) so that the saturation ratio of

water vapor with respect to ice ($S_{ice}$) was maintained at 1.50 at experimental temperatures of –20, –30, –40 ºC. This

$S_{ice}$ was chosen as it is a high but realistic saturation for the growth conditions in natural cirrus clouds (Comstock et

al., 2004; Dekoutsidis et al., 2023; Hoareau et al., 2016; Zhao and Shi, 2023) with a similarly realistic temperature range that still allows for high enough water vapor pressures and crystal growth rates to be experimentally viable.

For the gas stream mixing, this was first theoretically calculated as a dilution of saturated bubbler water vapor (19.42 g m$^{-3}$ at 23 ℃) with dry gas (< 0.03 g m$^{-3}$) to within 10% error. This was also then empirically determined by measuring the mixed gas dewpoint, calculating the water vapor pressure using the Sonntag parameterization and adjusting the flowrates accordingly.

The crystal growth rate ($J$) was calculated using a Fick diffusion term:

$$J = D_T \frac{C_0 - C_f}{l_m} \tag{2}$$

where $l_m$ is the thickness of the diffusion layer, $D_T$ is the diffusion coefficient of water vapor ($2.2 \times 10^{-5}$ m$^2$ s$^{-1}$) and $C_0$ and $C_f$ are the input and output water vapor concentrations. The value of $l_m$ is calculated by the Einstein equation:

$$l_m = \sqrt{2 D_T t_k} \tag{3}$$

where $t_k$ is the condensation time, i.e. residence time within the flowtube. $l_m$ was estimated to be 0.025 m, which is roughly the interior radius of the flowtube. $J$ is then multiplied by the interior surface area of the flowtube substrate (approximately 785 cm$^2$) to produce the total ice growth rate in the flowtube. While the crystal growth rate was theoretically calculated for all experiments, it was also empirically determined by dividing the sampled ice mass by the total experiment time.

With the experimental humidity determined and the flowrates fixed, the gas dilution factor of the bubbler gas is then also fixed. From this, the aqueous concentrations of all the analytes in the bubbler that produce the desired vapor concentrations in the flowtube can be determined. Using Henry's law and the compiled Henry's law constants and calculations from Sander (2023), (provided in Table S1 in the Supplemental Materials), the necessary aqueous concentrations to produce 10 ppbv of analyte vapor in the flowtube were determined. 10 ppbv was selected as the analyte vapor mixing ratio as it was a low mixing ratio that could still maintain signal in the ice samples. The Henry's law constant was first adjusted to ambient conditions using the equation:

$$H_T = H^\theta \cdot \exp\left(\frac{-\Delta_{sol}H}{R}\left(\frac{1}{T} - \frac{1}{T^\theta}\right)\right) \tag{4}$$

Here, Henry solubility ($H^\theta$) at the reference temperature ($T^\theta$) and the molar enthalpy of dissolution ($\Delta_{sol}H$) are used along with the gas constant (R) and ambient temperature ($T$) to correct to the Henry solubility ($H_T$). With the concentration/pressure defined $H_T$ adjusted to ambient conditions, the aqueous concentration of the analyte in the bubbler necessary to produce the desired amount of gas-phase analyte in the flowtube can be calculated using the equation:

$$[x]_{aq} = H_T^{cp} \cdot p_x^{FT} \cdot d \tag{5}$$

where the dilution factor ($d$) corrects the partial pressure of $x$ analyte in the flowtube ($p_x^{FT}$) to the partial pressure required in the bubbler due to the gas mixing dilution to provide the aqueous concentration ($[x]_{aq}$). These concentrations are stable in the bubbler, assuming that the mass fraction in the aqueous phase is much larger than the mass fraction in the vapor phase. The actual aqueous concentrations and Henry solubilities used can be found in

Table S1 the Supplementary Materials. While these calculations were performed to reach a target partial pressure of 10 ppbv, the actual partial pressure in the flowtube was determined by dividing the mass of analyte collected on the reagent-coated gas denuders and dividing by the total volume passed through the denuders. Corrections for mass error due to the breakthrough potential of a species through the denuder were made following the same method as (Kahnt et al., 2011). The average breakthrough potential for all species under these conditions was determined to be

less than 4%. While Kahnt et al. (2011) observed much higher breakthrough potentials than this at lower relative humidities than in these experiments, the absolute humidity in these experiments is lower by 5 orders of magnitude. Since water can both encourage and inhibit the derivatization, any changes in humidity conditions may alter the breakthrough potential of any of the analytes.

**2.3  Chemicals and Materials**

The derivatization reagent 2,4-dinitrophenylhydrazine (DNPH) was purchased from Sigma-Aldrich (~0.2 M, ~4% Phosphoric acid solution, Darmstadt, Germany). The denuder coating solution was prepared with 10 mM DNPH in acetonitrile (ACN). The following carbonyl compounds were obtained from Sigma-Aldrich (St. Louis, MO, USA): benzaldehyde (≥99%), methacrolein (95%), norcamphor (98%), (1R)-(+)-nopinone (98%), and methyl

vinyl ketone (MVK, with 0.5% hydroquinone and 0.1% acetic acid). Formaldehyde (30%, methanol-free) and

acetaldehyde (≥99%) were obtained from Roth (Karlsruhe, Germany). Hydroxyacetone (95%) and propionaldehyde

(97%) were obtained from Thermo Scientific (Darmstadt, Germany). Glyoxal (39% in water) and diacetyl (>98%)

were purchased from TCI (Toshima, Tokyo, Japan). Methylglyoxal (40% in water) was purchased from MP

Biomedicals (Irvine, CA, USA). d/l-camphor (97.5%) was obtained from WHI pharma services (Frankfurt,

Germany). These compounds were used without further purification. For the aqueous bubbler solution, 98 % LC/MS

grade water (Thermo Fisher Scientific) was used for the solvent and >99.8% technical grade nitrogen was used for

the carrier gas.

Supelco carbonyl-DNPH mix 13, a commercially available hydrazone standard solution, was purchased

from Sigma-Aldrich and used for the analysis of benzaldehyde, MVK, methacrolein, acetaldehyde, formaldehyde,

acetone, and propionaldehyde. Hydrazone crystals were prepared for benzaldehyde, nonpinone, norcamphor,

camphor, diacetyl, glyoxal, hydroxyacetone, and methylglyoxal. Benzaldehyde-DNPH was purified by

recrystallization from ethanol and then prepared in ACN. All other synthesized carbonyl-DNPHs were purified using

a solid phase extraction (SPE) method (Chromabond® $C_{18}$, 6 mL, 1000 mg bedweight) and then referenced to the

prepared benzaldehyde-DNPH standard with UHPLC-HRMS. This was then referenced to the Supelco standard.

The difference in the signal was about 23% (n = 6) between commercial and synthesized benzaldehyde-DNPHs

confirming the concentration of the benzaldehyde-DNPH. Since the uptake partitioning coefficients are unitless and

the sample matrices are the same, only relative quantitation is necessary for the calculation. This circumvents the

need for true quantitation as referenced to an external standard. However, since these standards are referenced to the

Supelco standard, true quantitation was performed for benzaldehyde, MVK, methacrolein, acetaldehyde,

formaldehyde, acetone, and propionaldehyde and pseudo-quantitation (estimating concentration by referencing

signal intensity to an internal standard) was performed for nonpinone, norcamphor, camphor, diacetyl, glyoxal,

hydroxyacetone, and methylglyoxal.

**2.4 Sample Collection and Preparation**

Three samples were collected from each experiment: input gas denuder extract, output gas denuder extract,

and ice. These samples were treated with a derivatization reagent and concentrated in preparation for analysis. The

gas denuders were prepared in the method described by Kahnt et al., (2011). Two 5-channel annular denuders with 750 mm length and 1 mm annular spacing (URG 4531, URG Corporation, Chapel Hill, NC, USA) were coated with XAD–4 resin following the method presented by Kahnt et al. (2011) and then coated with DNPH before immediate use. The resin coating was renewed after five experiments. At the conclusion of the experiment, the denuder samples were directly extracted three times with 50 mL of ACN by capping and inverting twenty times while rotating along its axis. These samples were then left overnight to ensure complete derivatization.

Ice samples were collected by methanol extraction of the flowtube. At the conclusion of the experiment, the flowtube and thermal exchange coil were sealed, disconnected from the setup, and taken into a walk-in cold chamber kept at –5 ℃. The caps were removed from the flowtube and the interior was rinsed with 14 mL of anhydrous methanol. The flowtube extract was weighed and then the water content was determined by measuring the refractive index using an Abbe refractometer. Knowing the percent water content (w/w%) of the extract and the total mass of the extract (g), the ice yield could be determined (g). This extract was then spiked with 0.1 mL of the DNPH solution and left overnight to ensure complete derivatization. This method prevents deposition by ambient humidity onto the flowtube substrate and is a more efficient recovery method than physical scraping, which was not a viable method due to the low ice masses deposited.

The samples were then all concentrated by rotary evaporation (25 ℃ at 150 mbar) and were reconstituted in 1 mL of methanol to be purified by SPE (Chromabond® $C_{18}$, 6 mL, 1000 mg bedweight). The cartridges were first flushed with 6 mL ACN, conditioned with 3 mL methanol and 6 mL of ultra-pure water. The denuder extract was loaded on the SPE cartridge and washed with 3 mL of methanol/water solution (5/95%, v/v%) to remove any phosphoric acid. The carbonyl-DNPHs were eluted using 10 mL ACN and stored out of light at –25 ℃ in a deep freezer. For analysis, 0.25 mL of the output gas denuder extract, 0.5 mL of the input gas denuder extract, and 1 mL of the ice sample were taken and evaporated to dry in a nitrogen evaporator at 18 ℃. These were reconstituted to 0.5 mL ACN/H2O (50/50, v/v%) for ultra-high-performance liquid chromatography coupled with high resolution mass spectrometry (UHPLC-HRMS). The dilution/concentration for these respective samples were performed to bring the expected concentration into quantitation range.

**2.5 UHPLC-HRMS Analysis**

Analysis was performed in triplicate using a Dionex UltiMate 3000 ultra-high-performance liquid

chromatography (UHPLC) system coupled to a heated electrospray ionization source (HESI) and a high-resolution

Q-Exactive Orbitrap mass spectrometer (HRMS) (all Thermo Fisher Scientific). A Hypersil Gold, C18, 50 x 2.0 mm

column with 1.9 µm particle size (Thermo Fisher Scientific) was used for the chromatography. Eluent A consisted of

98 % LC/MS grade water (Thermo Fisher Scientific) with 0.04 % formic acid and ACN (VWR Chemicals), eluent B

consisted of 98 % ACN and water, and the injection volume was 10 µL. Column temperature was held at 40 ºC. The

HESI source was used in negative mode, resulting in the formation of deprotonated molecular ions. Sheath gas and

auxiliary gas pressure was 40 and 20 a. u. (arbitrary unit) respectively. The temperature of the auxiliary gas heater

was 150 °C and the capillary temperature was 350 °C. The sprayer voltage was set to –4.00 kV. To further enhance

ionization, a post-column flow of 50 mmol $L^{-1}$ $NH_4OH$ in MeOH was added after 1 min at a flow rate of

0.1 mL $min^{-1}$. The following $H_2O$/ACN chromatography gradient was used: Starting with 30% B isocratically for 1

min, increasing to 80% at 10 min, then to 100% at 11 min, and back to 30% B at 11.5 min allowed to equilibrate to

initial conditions for 1 min. The first minute of eluent was ejected to waste to reduce excess unreacted DNPH being

fed into the HRMS. The mass traces used to identify the species in this experiment can be found in Table S2 in the

Supplementary Materials.

**2.6 Calculations**

The partitioning coefficient between the gas and ice phase K was calculated by the equation:

$$K = \frac{\rho_{ice} C_{ice}}{m_{ice} C_{gas}} \tag{6}$$

where $C_{ice}$ is the absolute mass of the analyte in ice (ng), $\rho_{ice}$ is the density of ice at the experimental

temperature (0.9194, 0.9200, 0.9208 g $cm^{-3}$), $C_{gas}$ is the concentration of the analyte in the gas phase (ng $m^{-3}$), and

$m_{ice}$ is the total mass of ice (g). For proper unit conversion, a factor of $10^6$ is applied for the conversion of $m^3$ to $cm^{-3}$. Practically speaking, the uptake coefficient $K$ can also be used as a sorption coefficient or a dimensionless uptake

coefficient with respect to the removal of trace gases in the upper atmosphere. A larger value for $K$ indicates more

uptake into the ice phase.

Strictly speaking, chemical uptake on growing ice crystals is a stationary state and not an equilibrium. However, diffusional crystal growth is slow relative to other modes of freezing (day vs. ms timescales) and so the

system can be approximated as an equilibrium state (Dominé and Thibert, 1996; Fries et al., 2007; Huffman and Snider, 2004). Then as a thermodynamic equilibrium, $K$ can also be used to calculate the Gibbs energy ($\Delta G$) of the uptake process at each temperature. This is a direct calculation using the equation:

$$\Delta G = -RTln(K) \tag{7}$$

where $T$ is temperature (K) and R is the ideal gas-constant (8.31447 J K$^{-1}$ mol$^{-1}$). These values describe the

thermodynamic potential of the uptake process. Positive values of $\Delta G$ indicate the analyte proceeds spontaneously to the gas phase while negative values of $\Delta G$ indicate the analyte proceeds spontaneously to the ice phase provided similar magnitudes of ice and gas volumes. Even lower, more negative values of $\Delta G$ would indicate more efficient uptake of the analyte into the ice phase provided the available ice volume is sufficient.

Continuing the thermodynamic analysis, the theoretical temperature dependence of the uptake coefficient $K$

—used in place of a sorption coefficient—can be determined with the van't Hoff equation, which when substituting with the Gibbs-Helmholtz equation produces the following:

$$\ln(K) = -\frac{\Delta H - T\Delta S}{RT} = -\frac{\Delta H}{R}\frac{1}{T} + \frac{\Delta S}{R} \tag{8}$$

where $\Delta H$ and $\Delta S$ are the heat of sorption and the sorption entropy respectively. Here the heat of uptake and uptake entropy are used instead. Performing a linear regression of $\ln(K)$ against $1/T$ using the van't Hoff equation

provides a slope of $-\Delta H/R$ and an intercept of $\Delta S/R$. Multiplying each of these values by $-R$ and R respectively produces the values of $\Delta H$ and $\Delta S$. Increasing values of the uptake enthalpy $\Delta H$ and the uptake entropy $\Delta S$ along with decreasing compound vapor pressure—i.e. the volatility of the pure analyte—can indicate that the uptake of the analyte is dependent on the physical parameters of the compounds and ice surface. Specifically, these are the parameters that determine thermodynamic sorption such as temperature, molecular mass, ice surface coverage,

surface morphology and porosity, surface crystallographic phases, quasi-liquid layer behavior, and crystal imperfections (Behr et al., 2006; Fries et al., 2006; Orem and Adamson, 1969; Sokolov and Abbatt, 2002). However, lower values of $\Delta H$ and poor linear regression can be a sign that the uptake process cannot be exclusively described by thermodynamic sorption.

## 3    Results and Discussion

### 3.1 Ice Crystal Growth

Nine ice samples were grown in the presence of vapor phase carbonyl compounds. While the target partial pressure for each species was 10 ppbv, the actual (excluding glyoxal) concentrations as referenced against denuder #1 was $11.5 \pm 2.5$ ppbv on average. The actual partial pressure for each species can be found in the Supplemental Materials. The typical ice yield at $-20$ $^\circ$C was calculated to be roughly 3.07 g while the actual yield was measured to be $5.77 \pm 0.45$ g on average. The corresponding theoretical and actual crystal growth rates were 128.1 mg hr$^{-1}$ and 237.8 mg hr$^{-1}$ at $1.55 \pm 0.14$ hPa water vapor pressure ($49.9 \pm 13.8$ % S wrt ice). For $-20$ $^\circ$C, the actual ice yield measurement is strictly an overestimate as the refractive index measurement was closest to the parabolic vertex of the water-MeOH mixture where small deviations in the refractive index produce large differences in the estimated water content. At $-30$ $^\circ$C, the calculated ice yield was 1.13 g while the average actual yield was $1.07 \pm 0.52$ g. The corresponding theoretical and actual crystal growth rates were 47.1 mg hr$^{-1}$ and 44.6 mg hr$^{-1}$ at $0.57 \pm 0.05$ hPa water vapor pressure ($49.6 \pm 14.1$ % S wrt ice). For $-40$ $^\circ$C, the calculated ice yield was 0.79 g while the actual was $0.57 \pm 0.11$ g. The corresponding theoretical and actual crystal growth rates were 16.4 mg hr$^{-1}$ and 11.8 mg hr$^{-1}$ at $0.20 \pm 0.01$ hPa water vapor pressure ($54.9 \pm 5.9$ % S wrt ice).

The lower actual ice yield than calculated is most likely due to deposition losses on non-extractible surfaces of the apparatus such as the caps of the flowtube or excess tubing inside the insulated chamber. Deposition of ambient humidity during sample extraction appears to be much lower than the losses present in the experimental setup. On average, the vapor deposition efficiency—that is the percent difference between the input and exhaust water vapor concentration, presumed to be the percentage of water deposited as ice—was 46%. This value never deviated more than 8% over the course of all experiments. This is potentially a geometric constraint of the flowtube apparatus as this value did not appear to change with temperature, flow rate, nor experiment time.

The size of any individual crystal was too small to reliably determine crystal morphology with nondestructive methods as the entire crystal yield was thinly coated over the interior surface of the flowtube (approximately 785 cm$^2$). This produced an ice coating that was typically less than 1 mg cm$^{-2}$ and often not evenly distributed across the surface of the flowtube. While true morphology could not rigorously be determined, ice deposits where crystal growth was quicker along with areas with needle-like structures were observed.

**3.2 Gas-Ice Partitioning Coefficients**

To establish background signal during uptake experiments, ice crystals were grown in the flowtube from pure LC/MS grade water in three separate experiments at –20 ºC without organic gases. The signals in the ice blanks and clean denuder extract were in the same range as analytical blanks, which were all below detection limits. This demonstrates that no measurable contamination occurred during crystal growth or sample extraction.

        At –20 ºC ($3.949 \times 10^{-3}$ K$^{-1}$), all compounds showed a $K < 1$. At lower temperatures, the $K$ for all
compounds increased except for MVK. This matches the expected behavior for exothermic deposition processes. Table 1 shows the calculated values for $K$ at each temperature while Figure 2 plots these values against each other as a van't Hoff plot. Values of $K > 1$ or $\ln(K) > 0$ indicate net uptake of the compound into the ice phase for a system with equal volumes of gas and ice. Conversely, values of $K < 1$ or $\ln(K) < 0$ indicate negligible uptake and that the compound favors remaining in the gas phase for a system with equal volumes of gas and ice. At –30 ºC ($4.112 \times 10^{-3}$ K$^{-1}$)
, formaldehyde is favorable to deposit to the ice phase while acetaldehyde reaches conditions where $K$ is close to 1 and the amount deposited to the ice phase and that remaining in the vapor phase are roughly equal. While formaldehyde is still mainly present in the gas phase at –20°C ($3.949 \times 10^{-3}$ K$^{-1}$), deposition of formaldehyde in the ice phase is favored from about –30°C ($4.112 \times 10^{-3}$ K$^{-1}$), while at this temperature acetaldehyde reaches conditions where $K$ is close to 1 and thus the amount deposited in the ice phase and the amount remaining in the gas phase are
approximately equal. At –40 ºC ($4.288 \times 10^{-3}$ K$^{-1}$), glyoxal and diacetyl also approach the point where $K$ is approximately equal to 1, while formaldehyde, acetaldehyde, acetone, and propionaldehyde preferentially deposit in the ice phase. Formaldehyde is the only species, however, that strongly favors the ice phase with a $K$ value in the order of $10^2$, while the other values are still around $10^0$ or far below. Formaldehyde has the highest ice-partitioning coefficients in this study; the potential reasons for this are discussed further in section 3.5. Formaldehyde is thought
to be the main source of OH radicals in the upper troposphere (Cooke et al., 2010; Fried et al., 2016). It then is likely that this ice uptake could be a significant influence on OH radical formation in the upper troposphere.

**Table 1. Average Gas-Ice partitioning coefficients from uptake experiments at different temperatures**

|  | $K$ at –20 ºC | $K$ at –30 ºC | $K$ at –40 ºC |
|---|---|---|---|
| MVK | $(2.21 \pm 1.23) \times 10^{-2}$ | $(2.05 \pm 2.08) \times 10^{-3}$ | $(9.01 \pm 8.26) \times 10^{-3}$ |
| Acetaldehyde | $(1.27 \pm 0.35) \times 10^{-1}$ | $0.98 \pm 1.37$ | $3.10 \pm 1.62$ |
| Acetone | $(1.87 \pm 0.54) \times 10^{-1}$ | $(2.92 \pm 3.84) \times 10^{-1}$ | $2.14 \pm 2.36$ |
| Benzaldehyde | $(7.68 \pm 2.86) \times 10^{-5}$ | $(6.64 \pm 4.98) \times 10^{-4}$ | $(1.43 \pm 1.14) \times 10^{-2}$ |
| Camphor | $(5.08 \pm 0.70) \times 10^{-4}$ | $(6.99 \pm 7.67) \times 10^{-3}$ | $(5.97 \pm 4.25) \times 10^{-2}$ |
| Diacetyl | $(3.69 \pm 1.13) \times 10^{-3}$ | $(3.17 \pm 2.94) \times 10^{-2}$ | $1.33 \pm 1.10$ |
| Formaldehyde | $(2.10 \pm 0.62) \times 10^{-1}$ | $(1.38 \pm 1.98) \times 10^{1}$ | $(1.03 \pm 0.63) \times 10^{2}$ |
| Glyoxal | $(9.01 \pm 1.22) \times 10^{-3}$ | $(6.78 \pm 2.45) \times 10^{-2}$ | $1.05 \pm 1.54$ |
| Hydroxyacetone | $(1.64 \pm 0.23) \times 10^{-3}$ | $(2.34 \pm 3.12) \times 10^{-2}$ | $(3.55 \pm 4.63) \times 10^{-1}$ |
| Methacrolein | $(3.94 \pm 1.89) \times 10^{-3}$ | $(1.15 \pm 1.43) \times 10^{-2}$ | $(2.10 \pm 3.06) \times 10^{-1}$ |
| Methylglyoxal | $(7.29 \pm 2.99) \times 10^{-3}$ | $(5.61 \pm 3.65) \times 10^{-2}$ | $(3.95 \pm 3.40) \times 10^{-1}$ |
| Nopinone | $(9.28 \pm 2.39) \times 10^{-6}$ | $(1.10 \pm 0.67) \times 10^{-4}$ | $(1.65 \pm 2.44) \times 10^{-3}$ |
| Norcamphor | $(4.32 \pm 2.07) \times 10^{-5}$ | $(9.05 \pm 2.72) \times 10^{-5}$ | $(6.15 \pm 7.59) \times 10^{-3}$ |
| Propionaldehyde | $(1.38 \pm 1.25) \times 10^{-1}$ | $(2.11 \pm 2.30) \times 10^{-1}$ | $2.93 \pm 3.23$ |

Every measured species except for MVK displays a strong correlation of ln(K) with inverse temperature and can likely be exclusively described by thermodynamic sorption. Furthermore, the uptake of these species can almost exclusively be attributed to codeposition with water vapor during crystal growth as sorption on nongrowing crystals has been demonstrated to be insignificant or completely reversible for almost all chemical species studied. This specifically includes acetone, acetaldehyde, formaldehyde, and benzaldehyde (Fries et al., 2006; Hudson et al.,

2002; Roth et al., 2004; Winkler et al., 2002). This observation of strong correlations with inverse temperature could indicate that K is controlled by transport, specifically if analyte transport is limited by accommodation at the ice-air interface (Davidovits et al., 2006; Jayne et al., 1991).

       MVK however shows a weak negative trend with inverse temperature with a nonsignificant correlation. The absence of this correlation for MVK complicates this view of K controlled by transport. However, this could be

explained by kinetic control resulting from transport phenomena occurring in either the gas or solid phases, i.e. processes that change the rates of transport of MVK relative to water vapor rather than a K that is controlled by an equilibrium established between MVK and ice. However, without sufficient evidence for a mechanism of kinetically controlled transport, the measurements of K here will be interpreted using equilibrium thermodynamics. While

outside the scope the current study, a kinetic explanation could be explored through theoretical calculations such as those found in Conklin and Bales (1993) or Reif (1965).

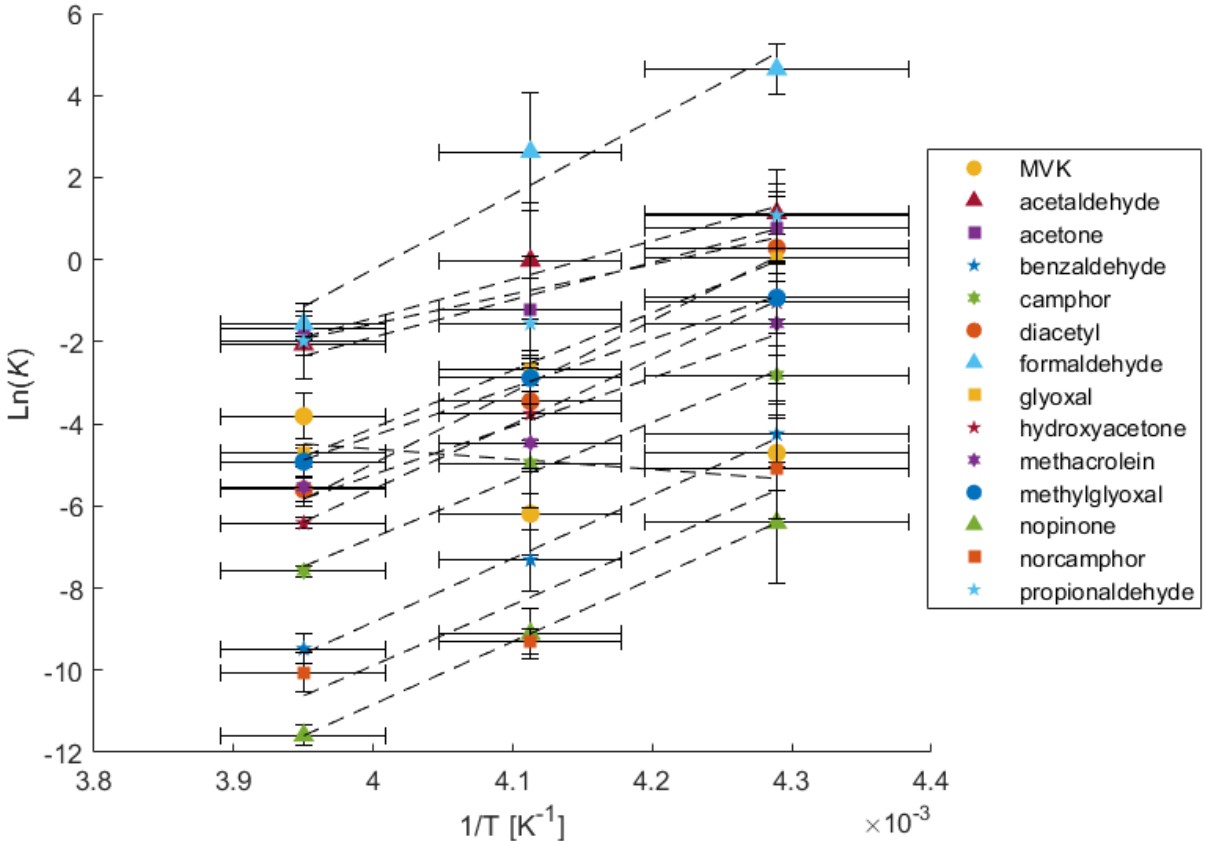

**Figure 2. Van't Hoff plot of inverse temperature ($K^{-1}$) against the natural log of the calculated partitioning coefficient $K$ (unitless)**

### 3.3 Thermodynamic Results

The Van't Hoff analysis of uptake can be used to estimate the enthalpy and entropy of thermodynamic sorption. Tables 2 and 3 apply the thermodynamic analysis from Eqs. 6 and 7 to the data given in Table 1 and the linear regressions seen in Figure 1. Table 2 provides the $\Delta G$ values produced by applying Eq. 6 the partitioning coefficients given in Table 1. Table 3 contains the slopes, intercepts, and regression coefficients ($r^2$) for the linear

regressions in Figure 1 as well as the calculated uptake enthalpy ($\Delta H$) and uptake entropy ($\Delta S$) produced by Eq. 7.

In Table 2, a $\Delta G > 0$ indicates unfavorable uptake of the species into the ice phase while $\Delta G < 0$ indicates favorable

uptake into the ice phase for a system with equal volumes of gas and ice. At –20 ºC, it is not favorable for uptake

into the ice phase to occur for any species. At –30 ºC, it is thermodynamically favorable for formaldehyde to be

taken into the ice phase. At –40 ºC, glyoxal, diacetyl, formaldehyde, acetaldehyde, acetone, and propionaldehyde are

thermodynamically favorable to be taken into the ice phase. Interpolating for these species, it becomes favorable to

deposit acetaldehyde, acetone, and propionaldehyde into the ice phase between –32 ºC and –35 ºC while

formaldehyde begins to deposit at –24.3 ºC. Both glyoxal and diacetyl deposit at approximately –40 ºC. Since cirrus

clouds often can occur at temperatures such as –60 ºC and lower, this data implies that uptake at those temperatures

could significantly affect the availability of these species. With these carbonyls being the typical source of

atmospheric OH radicals, this would significantly reduce the availability of OH radicals within cirrus clouds.

The other species—MVK, benzaldehyde, camphor, hydroxyacetone, methacrolein, methlyglyocal,

nopinone, and norcamphor—were unfavorable to deposit into the ice phase under any of the conditions studied here.

Extrapolating based on the linear regression in Table 3, the estimated temperature below which it is favorable to

deposit the species (excepting MVK) is presented in Table 2. For these species, they would presumably become

favorable to deposit into the ice phase within the range of –43 to –61 ºC, which is within the natural range for cirrus

clouds. With MVK having both a poor regression ($r^2 = 0.1235$) and nonexothermic behavior, the temperature at

which $\Delta G = 0$ cannot be meaningfully extrapolated.

**Table 2. Calculated $\Delta G$ of uptake at different temperatures.**

| | $\Delta G$ (kJ mol$^{-1}$) | | | Calculated $T$ where $\Delta G = 0$ |
|---|---|---|---|---|
| Temperature (°C) | –20 | –30 | –40 | (°C) |
| MVK | 8.0 | 12.5 | 9.1 | – |
| Acetaldehyde | 4.3 | 0.1 | –2.2 | –32.2 |
| Acetone | 3.5 | 2.5 | –1.5 | –35.4 |
| Benzaldehyde | 19.9 | 14.8 | 8.2 | –54.3 |
| Camphor | 16.0 | 10.0 | 5.5 | –49.9 |

| | | | | |
|---|---|---|---|---|
| Diacetyl | 11.8 | 7.0 | –0.6 | –39.6 |
| Formaldehyde | 3.3 | –5.3 | –9.0 | –24.3 |
| Glyoxal | 9.9 | 5.4 | –0.1 | –40.1 |
| Hydroxyacetone | 13.5 | 7.6 | 2.0 | –43.4 |
| Methacrolein | 11.7 | 9.0 | 3.0 | –47.3 |
| Methylglyoxal | 10.4 | 5.8 | 1.8 | –43.9 |
| Nopinone | 24.4 | 18.4 | 12.4 | –60.7 |
| Norcamphor | 21.2 | 18.8 | 9.9 | –56.5 |
| Propionaldehyde | 4.2 | 3.1 | –2.1 | –34.8 |

**Table 3. Slopes ($\Delta H/R$), intercepts ($\Delta S/R$), and regression coefficients ($r^2$) of regression lines from Figure 1, calculated uptake enthalpy ($\Delta H$), and uptake entropy ($\Delta S$).**

| | $-\Delta H/R$ (K) | $\Delta S/R$ | $r^2$ | $\Delta H$ (kJ mol$^{-1}$) | $\Delta S$ (J mol$^{-1}$ K$^{-1}$) |
|---|---|---|---|---|---|
| MVK | –2486.8 | 5.33 | 0.1235 | 20.68 | 44.4 |
| Acetaldehyde | 9392.4 | –38.99 | 0.9672 | –78.09 | –324.2 |
| Acetone | 7256.3 | –30.59 | 0.8972 | –60.33 | –254.4 |
| Benzaldehyde | 15452.5 | –70.64 | 0.9941 | –128.48 | –587.3 |
| Camphor | 14040.3 | –62.93 | 0.9934 | –116.74 | –523.2 |
| Diacetyl | 17426.7 | –74.67 | 0.9830 | –144.89 | –620.9 |
| Formaldehyde | 18197.2 | –73.02 | 0.9508 | –151.30 | –607.1 |
| Glyoxal | 14049.0 | –60.30 | 0.9960 | –116.81 | –501.3 |
| Hydroxyacetone | 15872.3 | –69.09 | 0.9997 | –131.97 | –574.4 |
| Methacrolein | 11804.2 | –52.45 | 0.9450 | –98.15 | –436.1 |
| Methylglyoxal | 11766.5 | –51.36 | 0.9987 | –97.83 | –427.0 |
| Nopinone | 15285.4 | –71.97 | 1.0000 | –127.09 | –598.4 |
| Norcamphor | 14765.6 | –68.95 | 0.8752 | –122.77 | –573.2 |
| Propionaldehyde | 9111.8 | –38.34 | 0.8690 | –75.76 | –318.7 |

All linear regressions calculated in Table 3 except for MVK have $r^2$ greater than 0.86 which indicate good linearity. The weakest regressions include norcamphor, propionaldehyde, and acetone which are between 0.869 and 0.897 while all the other regressions are above 0.945; those above 0.993 are benzaldehyde, camphor, glyoxal, hydroxyacetone, methylglyoxal, and nopinone. It can then be concluded that the gas-ice partitioning of all the species studied here except for MVK can be explained by the thermodynamic parameters of bulk uptake. The calculated $\Delta H$ and $\Delta S$ are then accurate descriptions of the thermodynamic process for these species' uptake in the ice phase. $\Delta H$ and $\Delta S$ are negative for all species except MVK and are within the ranges of –60 to –151 kJ mol$^{-1}$ and –254 to –621 J mol$^{-1}$ K$^{-1}$ respectively. These values indicate that at increasingly colder temperatures the uptake of all these species becomes more efficient and uptake decreases at warmer temperatures.

For MVK, since the linearity of the regression is poor, thermodynamic discussion of its measured partitioning coefficients is limited. Taking the calculated $\Delta H$ and $\Delta S$ at face value suggests that MVK behaves endothermically, and that uptake decreases at colder temperatures. This positive correlation with temperature has also been seen in the uptake of $C_1$–$C_4$-alkanols (Huffman and Snider, 2004), which is attributed to weakened water-water bonding when incorporated into ice due to supposed hydrogen bonding. MVK as a ketone is a hydrogen bond acceptor, as are the other ketones and aldehydes in this study, so this weak correlation is unlikely exclusively related to hydrogen bonding effects. More importantly, MVK is demonstrated to efficiently undergo functionalization and oligomerization in the aqueous phase through photooxidation (Renard et al., 2014). It is then a possibility that this photodegradation process is the main cause of the weak correlation that MVK has with inverse temperature. However, other similar compounds such as glyoxal and methacrolein also have demonstrated efficient functionalization and oligomerization through photooxidation but they do not exhibit the same poor regression as MVK. As an enone, MVK has also been observed to undergo unimolecular tautomerization, forming 2-hydroxybutadiene, however this involves high temperatures or intense UV (Couch et al., 2021). Additionally, almost all ketones are capable of tautomerization, not just MVK. Assuming that MVK's anomalous behavior is not due to its reactive properties obscuring observation, an explanation for this behavior could be found in its water-binding properties. While there are four observed conformational isomer adducts of MVK with water, the antiperiplanar conformation of MVK is preferred and is then stabilized by a network of two intermolecular interactions. These are the O − H⋯O hydrogen bond between the MVK oxygen and the water H atoms, and the second hydrogen bond C − H⋯O established between the water oxygen atom and one H atom from the methyl group (Cabezas et al., 2022).

If there's a change in the pattern of available hydrogen bonds, possibly incurred by the ice-growth kinetics that determine the principal facet, this could make MVK's preferred adduct conformation less favorable for accommodating to the ice surface. This would impose an energetic barrier to uptake which could explain MVK's anomalous behavior.

**3.4 Entropy-Enthalpy Compensation**

Plotting the calculated $\Delta H$ and $\Delta S$, as seen in Figure 3, demonstrates an apparent entropy-enthalpy compensation (EEC) effect, where $\Delta H$ scales proportionately with $\Delta S$. This relationship appears to also extend to the

455 calculated $\Delta H$ and $\Delta S$ for MVK. This form of linear correlation between $\Delta S$ and $\Delta H$, where it arises for a series of homologous compounds employed in a process, is referred to as the strong form of EEC (Sharp, 2001). This compensation effect appears to have good linearity with an $r^2$ of 0.9809 with no noticeable outliers with a compensation temperature (i.e. slope; $Tc = d\Delta H/d\Delta S$) of 235.5 K. EEC however has been known to arise as a statistical artefact with high $r^2$ and is often controversial (Grunwald and Steel, 1995; Krug et al., 1976; Leffler, 1955;

Leung et al., 2008; Liu and Guo, 2001; Moulik et al., 2019; Pan et al., 2015; Sharp, 2001). This is because methods such as van't Hoff analysis are indirect and do not measure $\Delta H$ and $\Delta S$ independently. Thus, for measurements on a limited temperature range, true observation of $\Delta H$ and $\Delta S$ can be obscured by trivial correlation arising from larger errors in determining $\Delta H$ than $\Delta G$ rather than some extra-thermodynamic mechanism of EEC (Sharp, 2001). Additionally, van't Hoff analysis assumes that the enthalpy change relative to the reference-state enthalpy change is

negligible (i.e., the heat capacity change is negligible). This may be invalid for an experimental temperature which may be significantly different from the reference-state enthalpy (Leung et al., 2008). Therefore, it is critical to evaluate EEC occurrence through statistical methods. For brevity, many of the specifics of this analysis has been moved to the Supplementary Materials under Section S1. The most relevant aspects and conclusions are presented here.

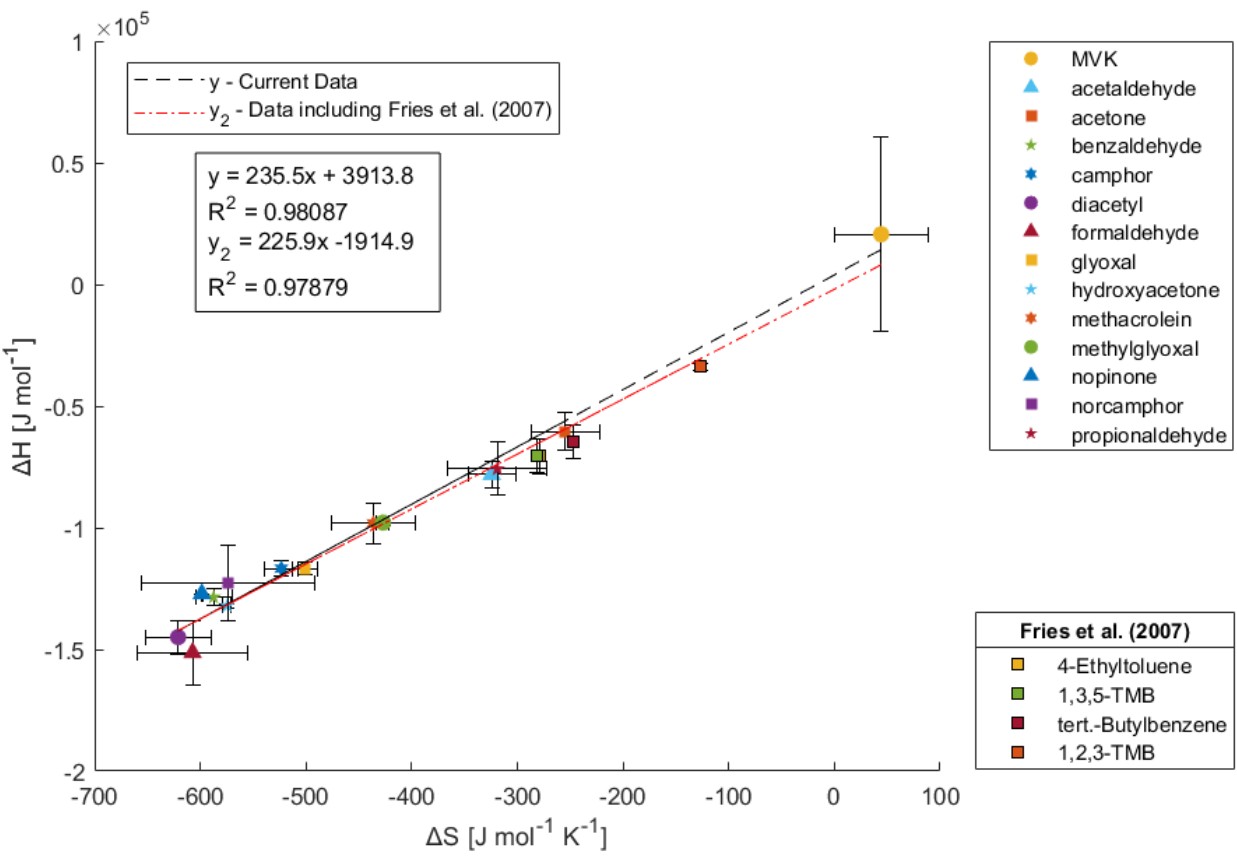

**Figure 3. Enthalpy versus entropy plot using the $\Delta H$ and $\Delta S$ determined from the Van 't Hoff plots in Figure 2**

To rigorously investigate EEC, the simple statistical verification of EEC provided by Griessen and Dam (2021) is applied to this data. The calculated dimensionless coalescence location parameter and Compensation Quality Factor pair ($k$, $CQF$) for this data is (–0.278, 0.062); which when plotted on Griessen and Dam's confidence contours, lies outside the 99% confidence contour for n = 12 (This dataset is n = 14). This indicates that there is 99% confidence that the EEC seen here is not of statistical origin. For good measure, if the data from Fries et al. (2007) is included in the ($k$, $CQF$) calculation, the calculated ($k$, $CQF$) pair is (–1.264, 0.205) and lies even further from the 99% confidence contour for n = 12 (This dataset is n = 18). From this statistical analysis, it can then be rigorously stated that the EEC seen in gas-ice partitioning is not artifactual in origin and likely has an extra-thermodynamic mechanism.

While an initial explanation of this mechanism might stem from a discussion on specific functional group-driven interactions, the appearance of this EEC effect includes aromatic hydrocarbons in addition to the ketones and aldehydes studied here. This could indicate that instead there may be weak, nonspecific supramolecular interactions. A few such explanations for EEC that are applicable to gas-ice equilibrium have been discussed in literature. Firstly, EEC due to solvation effects (i.e. solvent reorganization) are commonly discussed (Dragan et al., 2017; Leung et al., 2008; Lumry and Rajender, 1970; Pan et al., 2015). These discussions often center around the concept that any process that changes the free volume of nearby liquid water is inherently compensatory due to "structure making" and "structure breaking" of hydration shells. This explanation of EEC for gas-ice partitioning during depositional ice growth implies a surface liquid layer or quasi-liquid layer behavior. Indeed, there is already evidence for this as Huffman and Snider (2004) observe that at temperatures colder than approximately –20ºC there is overlap with models describing uptake into a surface liquid layer and previous uptake studies also use aqueous film models to account for $SO_2$ capture (Valdez et al., 1989).

Regardless of the specific mechanism of the EEC in this system, its presence does support (but does not necessarily prove) that there is a single source of additivity for the series of compounds studied (Lumry, 1995); i.e. a single thermodynamic component that controls the uptake process. Contrarily, there are also those who believe that EEC is not explainable and that it is an arbitrary phenomenon that arises from narrow free energy ranges (Moulik et al., 2019).

**3.5  Partitioning Coefficients versus Heat of Vaporization and Molar Mass**

The values for $K$ and ln$K$ at –20 ºC were regressed with several physiochemical properties, specifically molar mass (MM), HPLC retention time (RT), vapor pressure ($P_{vap}$ at 25ºC), heat of vaporization ($\Delta H^o_{vap}$) (Chickos et al., 1995), van der Waals volume (Zhao et al., 2003), and Henry solubility (Sander, 2023). These regressions were also made for $K$ and ln$K$ at –30 and –40 ºC, however all the notable trends are the same. Further, regressions were made with uptake $\Delta H$ and $\Delta G$, however the only significant correlations were from ln$K$ against ln$P_{vap}$, MM, $\Delta H^o_{vap}$, and van der Waals volume (n = 14, $r^2$ = 0.749, 0.737, 0.711, 0.702 respectively). ln$P_{vap}$ positively correlates while the rest of these properties all correlate negatively with ln$K$, indicating a relationship where larger compounds have lower uptake into ice and higher vapor pressures indicate more uptake. These correlations for MM and $\Delta H^o_{vap}$ at –20

ºC are displayed in Figure 4 while the same for –30 and –40 ºC are provided in the Supplemental Material as Figures S3 and S4. Additional correlations are provided in the Supplemental Material as Figures S5-S8. The higher $r^2$ for the regression against MM suggests that molecular size is the main contributor for uptake of carbonyls as opposed to

aqueous solubility or hydrogen bonding potential. The ability for ketones and aldehydes to hydrogen bond is limited as they are only capable of being bond acceptors. If hydrogen bonding between analyte and water played a significant role in the uptake process, then it would be expected that Henry solubilities would be more relevant contributors. Since carbonyls are unable to form hydrogen bonds between themselves, it's likely that the correlation from $\Delta H^o_{vap}$ is driven mostly from molecular size, as $\Delta H^o_{vap}$ is a property describing a pure substance. In this case,

MM and $\Delta H^o_{vap}$ positively correlate with an $r^2$ of 0.746 so their similar regressions are proxies of each other. Further, the residuals of the regressions for both properties are very similar. Almost all compounds stay on the same side of both regressions, i.e. few compounds change between positive and negative residuals. The exceptions to this are MVK, methylglyoxal, and hydroxyacetone. MM and van der Waals volume also positively correlate with an $r^2$ of 0.987, so van der Waals volume is considered a proxy for MM. $\ln P_{vap}$ however negatively correlates with MM with

an $r^2$ of 0.778, so $\ln P_{vap}$ can also be considered a collinear factor to MM.

The negative relationship of uptake with molecular size and positive relationship with vapor pressure is a unique finding that may seem counterintuitive if not considering inclusion into the ice lattice structure. One might expect that compounds with a higher affinity for the gas phase will remain in the gas phase and therefore have lower uptake coefficients. However, the reverse is observed. Compounds with higher vapor pressures and lower masses are

more readily taken into the ice phase. If in order to be taken into the ice phase, a compound must be incorporated into the ice crystal lattice structure, then this trend becomes more reasonable. Smaller compounds may induce less deviation in lattice structure relative to the preferred ice crystal structure. It may then be energetically less favorable for a larger compound to fit into the ice crystal as it forces a larger crystallographic defect. This trend might not be expected if analytes are phase separated from the ice crystal in grain boundaries. Other studies have suggested that

simple organics like formaldehyde are incorporated into the ice crystal volume under similar freezing conditions that produces snow (Perrier et al., 2002). This trend has been similarly hypothesized by Jost et al. (2017) for rime growth ice, but recent studies have not found this trend in rime growth ice nor bulk phase liquid freezing (Borchers et al., 2024; Gautam et a., 2024; Seymore et al., 2024). Huffman and Snider (2004) did not observe any specific dependence of uptake on compound saturation partial pressure nor molecular mass for acetone, touluene, or the $C_1$–

$C_3$ alkanols, but they observed a similar negative correlation with $\Delta H^o_{vap}$ as also seen here. Fries et al. (2007) did not

observe any trends between the physical properties of aromatic hydrocarbons and their uptake. An alternative

explanation could be that the preferential uptake of smaller compounds is due to possible mesoporous conditions on

the ice surface; i.e. pores that develop on the ice surface prevent the accommodation of larger compounds which are

sterically inhibited from entering small pores.


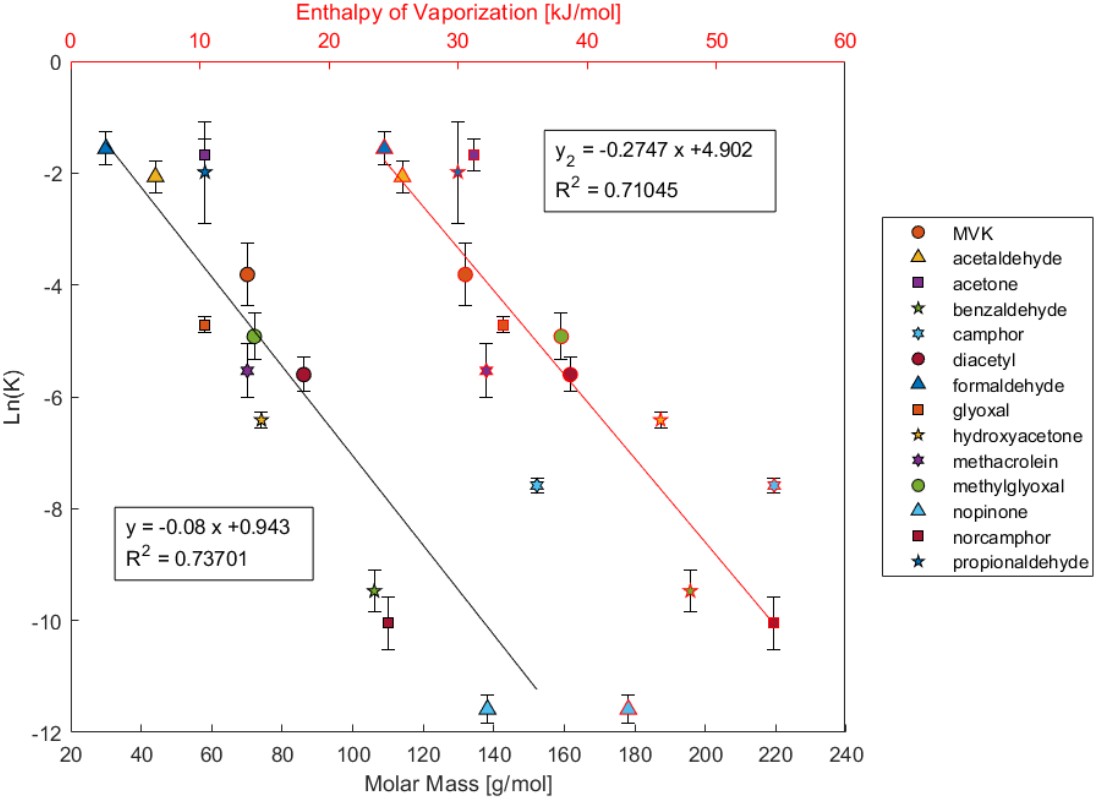

**Figure 4. Scatterplot of ln($K$) at –20 ºC versus the heat of vaporization (red line) and molar mass (black line).**

**4 Conclusions**

The uptake of carbonyls by ice crystals grown by deposition was studied at temperatures between –20 ºC

and –40 ºC. Ice was grown from the vapor phase in the presence of gas phase carbonyls using a flowtube apparatus

where ice saturation was controlled for a realistic saturation prevalent in in natural cirrus clouds. Uptake of the

carbonyl compounds from the gas phase during crystal growth was observed. Performing this experiment at

different temperatures allowed for the entropy and enthalpy of uptake to be determined. A linear correlation between $\Delta S$ and $\Delta H$ arose which was statistically validated and determined with 99% confidence that the EEC seen is not a statistical artifact. This compensation behavior could be an indication of a surface liquid layer or quasi-liquid layer behavior involved in this process; or it could also indicate a single dominant influence on a compound's uptake. The most significant chemical properties correlated with uptake were identified to be molecular size and vapor pressure. Their relationship to uptake is indicative of incorporation into the ice crystal structure, as smaller compounds have higher observed uptakes.

In agreement with previous studies, these results indicate that deposition into the ice phase is a possible uptake process for organic compounds in cirrus clouds. Since ice growth in the troposphere at low temperatures is a major contributor to precipitation, in-cloud scavenging is a plausible explanation for the occurrence of organics in fresh snow (Roth et al., 2004; Su et al., 2021). Therefore, interactions of ice and organic compounds can influence atmospheric transport and removal of those compounds from the atmosphere by deposition. The partitioning coefficients observed here are however below—by at least 3 orders of magnitude—the 10 mol m$^{-3}$ Pa$^{-1}$ threshold from Crutzen and Lawrence (2000) to be considered a substantial atmospheric removal process. Importantly, the partitioning of these compounds to the ice phase are governed in part by the volume ratio of the partitioning phases and may be less relevant in situations where the amount of ice is magnitudes smaller than the amount of gas, which describes nearly all natural cloud conditions (Brimblecombe and Dawson, 1984). Specifically, to appreciably accumulate a significant fraction of a compound from the gas phase, a large K is needed in order to overcome the difference in ice and air, likely a $\ln(K) > 10$.

Compared to the well-studied uptake of HNO$_3$, the uptake of most species studied here are 2-3 orders of magnitude lower while acetaldehyde, acetone, formaldehyde, and propionaldehyde may be considered in comparable range (Von Kuhlmann and Lawrence, 2006; Zondlo et al., 1997). Notably, as these species are relevant as significant sources of OH radicals in the upper troposphere (Cooke et al., 2010; Fried et al., 2016), it then is likely that this ice uptake could be a significant influence on OH radical formation in the upper troposphere at temperatures less than the range studied here, notably –60 ºC where the extrapolated uptake could reach $\ln(K) > 10$.

For comparative reference, the partitioning coefficients reported here can be converted from dimensionless coefficients to mol m$^{-3}$ Pa$^{-1}$ by dividing by $RT$, which is provided in the supplement. Further, the uptake coefficients

for carbonyls are on average smaller than those for aromatic hydrocarbons studied by (Fries et al., 2007), which have already been estimated to be removed primarily by photochemical processes rather than ice phase scavenging. While this removal process cannot be considered substantial in terms of mass transport, it may be relevant as an influence on vertical tracer transport or OH radical formation.

These measurements are exclusively a description of the gas to ice solid solution equilibrium and neglect investigation of gas to liquid or liquid to ice equilibrium. Critically, the measurements here could be influenced by liquid water condensation within the Flowtube. Provided some regions of the glass substrate are clean and free from ice nucleating sites, some liquid phase water could condense directly to the substrate and then freeze thus competing with the diffusional growth ice. Additionally, some liquid water could condense into droplets within the gas stream and then impact on the substrate thereby contributing to the deposited ice via riming. These alternate modes of
freezing are unlikely to occur under these conditions as the clean carrier/dilution gas has virtually no nucleating particles for liquid droplet condensation and the glass Flowtube substrate is ideal for facilitating dendritic ice crystal growth (Chen et al., 2020). However, the presence of liquid water condensation within the Flowtube could be an alternative explanation to the observed exothermic uptake trends, assuming that Henry's law uptake into the liquid phase increases with decreasing temperature below 273 K. If this were the case, Henry solubilities should be the
dominate factor controlling uptake and it would also be expected that more water-soluble species like glyoxal would have much higher uptake coefficients than less water-soluble species like formaldehyde. Neither of these are observed here so it can be concluded that liquid water condensation within the Flowtube is unlikely to have significantly influenced the present measurements.

    These partitioning coefficients also do not directly describe whether a compound is actually incorporated
into the ice crystal lattice or if it phase separates into crystal grain boundaries, but only its uptake into the bulk phase. However, the negative correlation of molecular size and uptake may suggest incorporation into the ice crystal lattice or void space. It is also difficult to say if this data describes uptake into a liquid solution phase that coexists with ice, but the observed compensation effect may insinuate its presence. While the measurements here are for multicomponent mixtures of compounds, single component uptake is likely the same, which is also supported by
Huffman and Snider (2004).

Further investigation to determine the contribution of liquid layer influence should focus on measuring on ice-specific surface area and the volume of solution associated with the liquid layer. Additionally, similar experiments with other families of compounds are required to better understand the root of the compensation effect seen here. Crystallographic analysis of this data may also yield more information about the ice uptake process. With more investigation to reveal the main contributors of additivity, it seems possible that the EEC seen here could be used to help generalize descriptions of the uptake process in models.

**Acknowledgements**

This work was funded by the Deutsche Forschungsgemeinschaft (DFG, German Research Foundation) – TRR 301 – Project-ID 428312742.

This work was supported by the Max Planck Graduate Center with the Johannes Gutenberg University of Mainz (MPGC) as well as by internal funding from the Max Planck Institute for Chemistry (MPIC). We would also like to acknowledge the mechanical workshop of the Johannes Gutenberg University Institute of Atmospheric Physics (JGU-IPA) and the glass workshop of the Max Planck Institute for Polymer Research (MPIP) for their technical expertise and contributions.

Special thanks to Jan Wallner for his contributions testing the experimental setup.

**Author Contributions**

JS, MS, AT, SM participated in designing the experiments; JS, AT, SM participated in constructing the experimental apparatus; JS prepared the solutions for experiments, performed the experiments, and collected the samples; JS, CB conducted the analytical measurements; JS analyzed the data and wrote the manuscript draft; JS, MS, AT, CB, TH reviewed and edited the manuscript.

**Competing Interests**

The contact author has declared that none of the authors has any competing interests

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
