# Peer review of "Volume Uptake of Carbonyls during Diffusional Ice Crystal Growth"

_EGUsphere, 2025_

## Referee Comment (RC2)

**"Gas-ice partitioning coefficients of carbonyls during diffusional ice crystal growth"**

**By J. Seymore et al. (2025) for EGUsphere**

Reviewed by Jeff Snider, University of Wyoming

**Overall**

The authors describe a laboratory setup and use it to characterize the amount of organic vapor
incorporated into ice grown by deposition. This reviewer participated in a similar study ~ 20 years ago.
These are not easy measurements to make. Interpreting the results is also difficult.

One of my major critiques concerns the tentative conclusion that a surface layer is the "single dominant
influence" on uptake. This runs counter to the discussion of how uptake might be controlled by
accommodation or affinity within bulk ice.

The authors should consider my critiques and reply with a revised manuscript.

**Major Critiques:**

On L276, you say that "Positive values of DG indicate the analyte favors the gas phase while negative values
of DG indicate the analyte favors the ice phase.  Even lower, more negative values of DG would indicate
more efficient uptake of the analyte into the ice phase."  Also, on L330, you say "Values of K > 1....indicate
net uptake of the compound into the ice phase. Conversely, values of K<1...indicate negligible uptake and
that the compound favors remaining in the gas phase." And, on L376, you say that "...DG>0 indicates
unfavorable uptake of the species into the ice phase while DG<0  indicates favorable uptake in the ice
phase."

As I demonstrate below, your statements are illogical for a system with an amount of ice which is
magnitudes smaller than the amount of gas. I will symbolize these amounts as volumes (Vs and Vg) and will
insist that Vg/Vs >> 1. Also, I note that Vg>>Vs is the situation within clouds.  With these constraints, and
assuming sufficient time for equilibration, the K you report allows for determination of analyte amount (as a
mole count) within the solid (KCgVs), within the gas (CgVg), and the fraction of analyte within the solid.

Fraction of analyte within the solid = KCgVs / (KCgVs+ CgVg) = K / (K + Vg/Vs)

My point is this: Since K is not large compared to 1 (Fig. 4), and Vg/Vs >> 1 is the situation within clouds, it
does not make sense to say that "negative values of DG indicate the analyte favors the ice phase", or that
"Values of K > 1....indicate net uptake of the compound into the ice phase", or that "DG<0  indicates
favorable uptake in the ice phase."  Also, while it is logical to say that "Positive values of DG indicate the
analyte favors the gas phase", and that "Conversely, values of K<1...indicate negligible uptake and that the
compound favors remaining in the gas phase",  and "DG>0 indicates unfavorable uptake of the species into
the ice phase", these last statements are also true for a K that is magnitudes larger than 1.

In summary, you need to rewrite these sections of text.  A suggestion:  Think of DG is a placeholder for K,
and not as an indicator of how uptake alters the partitioning of the analyte within clouds. If you do not want
to focus on clouds please recognize that it is an unusual experimental setup that allows for Vg/Vs ~ 1 while
also assuring time for equilibration.

Related to the previous comments and critique:

1) Since partitioning is strongly on the side of gas-phase, I don't accept your assertion that uptake into ice
"...may be relevant as an influence on vertical tracer transport" in the Conclusion.

2) My formulation of partitioning comes from Brimblecombe and Dawson
(https://doi.org/10.1007/BF00127265).  As far as I can tell, they were the first to put the concept into the
literature.

3) There is a treatment of partitioning in the atmospheric chemistry textbooks.  One of these is Lamb and
Verlinda (Physics and Chemistry of Clouds, Cambridge University Press, 2011; see pp. 166-168).

4) I recommend that you report your ice uptake coefficient with dimension of mol m-3 Pa-1. This is your
framework for the Henry Law discussion (Equations 5 and 6).  With this change you can eliminate a
confusing sentence (L210-L211). It would also eliminate the need for a translation from a dimensionless K
to dimensional K, in the Conclusion.

L134 says that $S_i$ = 1.5 was a constraint in all experiments. This implies that either deposition or
condensation can occur. I checked my assertion here:

| T,[K] | Si,[-] | ei(T),[Pa] | e,[Pa] | es(T),[Pa] | RH,[%] |
    |---|---|---|---|---|---|
| 253.15 | 1.50 | 103. | 155. | 125. | 123. |
| 243.15 | 1.50 | 38. | 57. | 51. | 112. |
| 233.15 | 1.50 | 13. | 19. | 19. | 101. |

Note that the relative humidity (relative to saturation over liquid water) exceeds 100 % at all temperatures (-
20, -30, and -40 °C). Since you are using "tank" air, it is unlikely that droplets formed, impacted, and
contributed to the ice deposit via riming.

I recommend that you consider the following as a process that may have occurred in your experimental
setup:

Provided some regions on the glass are clean (i.e., no ice nucleating particles) you may be condensing
liquid directly from the vapor. Freezing would then occur, for example, once a liquid domain touches an ice
domain. The latter could have been previously frozen or previously deposited.  In either case, the existence
of liquid water would rationalize your exothermic uptake coefficients. Here, I'm assuming that Henry's law
uptake (into the liquid) increases with decreasing temperature below 273 K.

Related Recommendations:

1) I recommend that you refer to your measurements as an uptake coefficients, as sorption coefficients, or
as an ice-gas partitioning coefficients.  You complicate the reading by using all of these. The place to do
this is in the Introduction not in Section 2.6.

2) Since the cloud chemistry community uses "partitioning" to describe analyte mass continuity, within an
element of cloud, I recommend that you not use partitioning to modify the coefficient you are evaluating.
Similarly, I encourage that you do not use "partitioning" as a place holder for the process you are
investigating.

3) I do not think you need Equation 2. There are tables of saturation pressure in the textbooks.  E.g., Rogers
and Yau, Third Edition, Elsevier, 1989; Table 2.1.  The important thing is that you check your Equation 2 for
dimensional consistency and for adequate numerical precision compared to tabulated data.

4) There is no reason to report the gas constant with six-digit precision or to report the reciprocal
temperatures (p. 14) with four-digit precision.

5) The density of ice can be taken to be a constant.

Your discussion on L175-L187 should be revised.  You are using aqueous solutions to generate a prescribed
amount of gas-phase analyte within the flowtube. The $[X]_{aq}$ (Equation 6) is what's required to produce a
specified gas-phase mixing ratio (10 ppbv). So, I recommend that you put the target gas-phase mole
fraction ($10.x10^{-9}$), and your pressure, into Equation 6.

Please note:

L179. You are producing a partial pressure, within the flowtube, not a "gas concentration."  This should be
corrected here and elsewhere.

It's not clear why you have the superscript "cp" in Equation 6. Also, it's quite common to use Kh, not H, for
the Henry solubility, at least in the cloud chemistry community.

About "breakthrough."  I don't understand your statement that Kahnt et al. (2011) observed larger
breakthrough, at lower relative humidities, but your absolute humidity is much lower. Do you mean that
Kahnt et al. (2011) observed larger breakthrough at _low_ relative humidities?  Additionally, relative
humidity and absolute humidity are related, but it is not clear how to compare your humidity condition
(absolute) to the humidity condition in Kahnt et al. (2011) (relative).

Please note the title of Kahnt et al. (2011): "Denuder sampling techniques for the determination
of...derivatization methods."  There are places where you use the modifier "derivation" instead of
"derivatization." Please check throughout.

The word "massed" is used.  Do you mean weighed?  It seems that the mass of the ice was derived by
weighing together with additional information.  On L225-233 you say that the flow tube was rinsed
(methanol), that the extract was "massed", and that the solution was evaluated in a refractometer. The
acquired information is sufficient for determining the mass of $H_2O$ that was extracted from the tube.

1) Please correct/change all instances of "massed".

2) You refer to the "collected ice mass" on L159.  Should this be "sampled ice mass" or "extracted sample
mass"?  There are other instances of "ice mass". You should consider changing these, for clarity.

3) On L305-L309, you report the calculated ice mass (Equation 3) and you compare to the collected ice
mass (aka, the "yield."). In your procedure, it seems, the methanol does not capture all the ice. Hence, the
expectation is calculated > measured. Looking at data from the -30 and -40 $^\circ$C, that expectation is verified.
In our paper (Huffman and Snider 2004), a calculation overpredicts the observed ice mass. Splintering of
the ice sample, during the uptake experiments, was suspected.  In your experiments, do you have evidence
of splintering?

4) Your visual observations (L318-319) are contradictory.  The coatings were "either over the entire surface"
or "not evenly distributed."  Which of these best represents what you saw?

5) All one can visualize is a macroscopic ice deposit, not the (microscopic) nucleation sites. Accordingly, I
recommend that you revise L321.

Contamination during the processing of the samples (the $H_2O$ samples and the cartridge samples) is a
possibility. You addressed this on L325-L328. Your approach is to process blank ice samples grown from
liquid.

1) Why not collect/process "blank" ice samples taken from the flowtube when it's operated with no
analyte?

2) You say that the blank signals were in the same range as analytical blanks, which were all below
detection limit. It's my opinion that you should tabulate the analytical blank values - or tabulate their
average and standard deviation – and tabulate the same for the blanks ice samples.

I may have missed this. Did you do an analysis of analyte amount in denuder #1, denuder #2, and in the
flowtube and use that information to quantify error in K coming exclusively from the chemical analysis
method?  This seems like a useful thing to do....mass must be conserved.

L39 - Neither Gautam et al. (2025)  Seymore et al. (2025) are in the bibliography.

L502 – Neither Gautam et al. (2024)  Seymore et al. (2024) are in the bibliography.

Regarding your tentative conclusion that a surface layer is the "single dominant influence" on uptake.  This
was examined by Valdez et al. (https://doi.org/10.1029/JD094iD01p0109). This reference is not in your
bibliography.

**Less-than-major Corrections and Less-than-major Critiques:**

"ACN", define where this acronym is first used.

L144. Remove "using the hygrometer."

L150. Remove "which is equivalent to the diffusive flux."

L153. A cloud physics textbook (Rogers and Yau; Third Edition, Elsevier, 1989) says that the diffusivity is
20% smaller at -40 °C compared to -20 °C.

L157. Please see my earlier comment about condensation and deposition. You are interpreting $t_k$ as the
time for "deposition", correct?

L202-L204. Is "hydrazone" different from "hydrazine" in DNPH?

L262-L267. I recommend that you not use "Cice" for the absolute mass of analyte in ice. Also, is the
modifier "absolute" implying something? Recommend that you remove "absolute."

L275 You say: "These values describe the energy available for the uptake process." I do not agree with this.
Think of DG is a placeholder for K. Also see my first critique.

L361 "...and water". Do you mean "...and ice"?

L411 "Endothermically" or "nonexothermically"? Be consistent.

L416-L417 I don't understand how photodegradation of MVK, within the atmosphere, can be the cause of
the weak correlation with inverse temperature, in your laboratory study.

L430 "artifactully". Is there a better way to say this?

L435 The letter "H" is being used to represent several properties. There is the Henry solubility (Equation 5),
the enthalpy of solution (BTW, some refer to this as the enthalpy of gas-to-liquid transfer), and the enthalpy
change you derive via the van't Hoff regression. You can avoid introducing another "H." Do that by
acknowledging that the enthalpy change you derive for a rather cold experimental temperature may be
significantly different from the reference-state enthalpy. Related to this, why not represent the Henry
Solubility (Equation 5) with Kh? That is common, and it would eliminate one of the "H" symbols.

L480 Which "solubility"? I think you are referring to the Henry Law solubility, but I'm not sure.

L456 Is this the first occurrence of "additivity"? It is not clear what you are implying by "additivity"

L457 I recommend something like this: "...could help to generalize descriptions of the uptake process in
models"

L539 It is not clear what you are implying by "..void space."

L711 Check the format of the Warhaft reference.

---

## Author Comment (AC1)

We would like to thank the Editorial Support team for giving us the opportunity to respond to the reviewers' comments in open discussion. We greatly appreciate the reviewer's careful reading and review of this manuscript. We have addressed each reviewer's comments and suggestions, as shown in our responses below. The lines, figures, and sections referenced match the numbering in the version of the manuscript presently uploaded for discussion. A revised version of the manuscript will be uploaded for the handling editor's consideration.

**Reviewer(s)' Comments to Author:**

**RC1**:

**The paper describes laboratory measurements of gas-ice partitioning coefficients for 14 carbonyls along with an analysis of thermodynamic properties. The paper provides very useful information of these partitioning coefficients that can be applied in atmospheric chemistry models. The analysis is interesting, indicating a quasi-liquid surface layer may play a role, but also that carbonyls with lower molecular mass are more likely to be taken up into the ice crystal lattice. Scientifically, the paper is good. Its analysis brings up a number of questions, including why methyl vinyl ketone (MVK) behaves differently than the other carbonyls.**

**The paper does need some improvement. In general, there is a need for better clarity: explaining the methods and analysis for those less familiar with these tools and explaining the results in relation to atmospheric chemistry implications. As noted, the MVK behavior is curious. It may be worth having a separate small section synthesizing what was learned about MVK with possible explanations as to what causes its behavior and potential future areas of investigation.**

**Below I list several comments that I would like the authors to address before considering the paper for publication.**

**Major Comments**

1. **The results for MVK are perplexing for both their weak response to temperature but also the MVK partitioning coefficient values are much different than those for methacrolein, which has the same molecular mass. I was curious whether the functional group(s) of the carbonyl play a role in its deposition onto ice. At first glance, the aldehydes seem to have higher partitioning coefficients than the ketones, but that is likely because the aldehydes generally have a lower molecular mass. Further, another pair, acetone and propionaldehyde, appear to**

**have similar partitioning coefficients with each other. Nevertheless, the results in Figure 4 show partitioning coefficients varying by two orders of magnitude for the same molecular mass (e.g., 58 g/mol). Could the authors comment on the role of functional groups and suggest other properties that may be controlling the carbonyl's ice partitioning coefficient in the manuscript?**

**The explanation of the weak response of MVK with temperature is a bit speculative. If it is the kinetic control of transport, then could this be investigated from theoretical calculations? If doing additional calculations is beyond the scope of the current paper, then recommendations for further research should be stated.**

We discuss some of the potential reasons for MVK's anomalous behavior at the end of section 3.3 but we have expanded this discussion to include more postulation on functional group influence. We do think however that the more relevant properties to be molar mass based on our discussion in 3.5.

> As an enone, MVK has also been observed to undergo unimolecular tautomerization, forming 2-hydroxybutadiene, however this involves high temperatures or intense UV (Couch et al., 2021). Additionally, almost all ketones are capable of tautomerization, not just MVK. Assuming that MVK's anomalous behavior is not due to its reactive properties obscuring observation, an explanation for this behavior could be found in its water-binding properties. While there are four observed conformational isomer adducts of MVK with water, the antiperiplanar conformation of MVK is preferred and is then stabilized by a network of two intermolecular interactions. These are the $O-H\cdots O$ hydrogen bond between the MVK oxygen and the water H atoms, and the second hydrogen bond $C-H\cdots O$ established between the water oxygen atom and one H atom from the methyl group (Cabezas et al., 2022). If there's a change in the pattern of available hydrogen bonds, possibly incurred by the ice-growth kinetics that determine the principal facet, this could make MVK's preferred adduct conformation less favorable for accommodating to the ice surface. This would impose an energetic barrier to uptake which could explain MVK's anomalous behavior.

We have added additional recommendations for research into theoretical calculations investigating kinetic control of transport.

> While outside the scope the current study, a kinetic explanation could be explored through theoretical calculations such as those found in Conklin and Bales (1993) or Reif (1965).

2. **The paper would benefit by revising the text so that it is more easily understood by those who are not very familiar with laboratory studies. That is, explaining the experiments in plain English as well as providing the technical details would be useful. This should be applied to the analysis approach and discussion as well.**

We have added a more simplified summary description of the experiments in section 2.1. We have also added more simplified summary descriptions of the analysis in section 2.4.

> In the present experiment, ice crystals are grown in the presence of carbonyl vapors. Their concentrations in the ice-phase and the gas phase are then measured to determine their uptake coefficients.

> Three samples were collected from each experiment: input gas denuder extract, output gas denuder extract, and ice. These samples were treated with a derivatization reagent and concentrated in preparation for analysis

**Specific Science Comments**

1. **I am confused by the remark in this manuscript that states, "The gas-ice partitioning coefficients observed here are below the 10 mol m$^{-3}$ Pa$^{-1}$ threshold given by Crutzen and Lawrence (2000) to be considered a substantial atmospheric removal process." (line 25-26 abstract and line 528). The threshold given by Crutzen and Lawrence is the trace gas solubility, which they used as an indicator for scavenging by cloud particles. However, what I learned from this current manuscript is that the solubility of the trace gas did not play a role in its direct uptake onto ice. For example, glyoxal is very soluble (much more than formaldehyde) but its ice uptake is smaller than that of formaldehyde. I do not see the relevance of making this remark.**

The comparison to the trace gas solubility is not incongruous. As the reviewer mentions, this threshold is a cloud scavenging term. To be more specific, the threshold by Crutzen and Lawrence (2000) is not strictly a trace gas solubility, it is a volume uptake coefficient. The ice-gas partitioning coefficients we present are also volume uptake coefficients and indicators of scavenging by ice particles. This comparison is also made by other studies in the literature that present ice-gas partitioning coefficients, such as Huffman and Snider (2004) and Fries et al. (2007).

2. **The 14 carbonyls investigated in this study were never introduced individually. On line 74, it states that 14 carbonyls are investigated and then the reader does not learn which carbonyls until section 2.3 which discusses the materials used. In the Introduction, it would be good to have a paragraph listing the carbonyls providing explanations of their atmospheric chemistry relevance, especially for the upper troposphere where the impact of deposition onto ice will have its greatest effect.**

We have expanded the discussion in Section 1.1, introducing the selected carbonyls more individually and more thoroughly presenting carbonyl participation in ozone and SOA formation.

> Carbonyls as a class of trace atmospheric constituents are highly relevant secondary organic aerosol (SOA) precursors and intermediates (Ervens & Kreidenweis, 2007; Galeazzo et al., 2024; Srivastava et al., 2022; Yu et al., 2014). They are ubiquitous and play vital roles in tropospheric photochemistry and oxidative capacity, which affects radical cycling and ozone formation (Xu et al., 2023). Specifically, the photolysis of carbonyls is an important source of peroxy radicals in the atmosphere. After photolysis, the aldehydic group (-CHO) decomposes and forms the $HO_2$ radical with the addition of $O_2$. This peroxy radical is then a source of atomic oxygen for ozone formation (Q. Liu et al., 2022). Formaldehyde, acetaldehyde, and acetone are typically the most abundant and considered the main contributors to ·OH reactivity and dominate the ozone formation potential of the total oxygenated volatile organic compounds (VOC), while less abundant species like benzaldehyde make a minor contribution to the ·OH removal rate and inhibit ozone formation (J. Wang et al., 2022). Additionally, lower vapor pressure carbonyls can be oxidized to form SOA through gas-particle partitioning. This chemical evolution and physical transformation results in a range of lifetimes for various carbonyls from hours (e.g., below 1 hr for unsaturated aldehydes against OH oxidation, 1–3 hr lifetime for glyoxal against photolysis and OH oxidation, 1.3 hr for formaldehyde and glyoxal under overhead sun conditions) to a few days (i.e., 15 days for acetone with respect to the oxidation by OH) depending on their structures (Jacob, 2021). Compounds like glyoxal and methylglyoxal are significant contributors to SOA formation via aqueous-phase chemistry (Ling et al., 2020) and many other such SOA contributing carbonyls are isoprene products, like methacrolein, methyl vinyl ketone (MVK), hydroxyacetaldehyde, or hydroxyacetone (Grosjean et al., 1993). Nopinone, diacetyl, camphor, norcamphor, and propionaldehyde are other naturally occurring carbonyls of interest that present a variety of structures and properties for comparison. Notably nopinone, camphor, norcamphor, and propionaldehyde are analogous bicyclic monoterpenoids while diacetyl is another diketone similar to glyoxal and methylglyoxal.

3. **Lines 40-44. These sentences are still quite general in explaining how carbonyls are relevant to atmospheric composition and chemistry. There should be statements about their role in ozone formation as a source of peroxy radicals when they photodissociate and a better description of how they contribute to secondary organic aerosol formation.**

We have included more discussion on the role of carbonyl participation in ozone and SOA formation in the introduction, which is presented in the previous response.

4. **Lines 44-48. If a trace gas is being "removed from the atmosphere" by either dry or wet deposition, then would not one want to determine the resultant deposition? Further, the results of this paper show that deposition of carbonyls onto ice is greater at colder temperatures relevant to the upper troposphere. Specifically, this process would be happening in cirrus clouds and convective anvils. The carbonyls are moved from the gas phase to the ice phase. However, at some point the ice will sublimate or fall and melt. When that happens, the carbonyls will be released back to the gas phase. In my mind, this carbonyl deposition onto ice must be characterized as an effect on tropospheric gas-phase chemistry while the ice cloud is present, and then as a source to the local region for gas-phase chemistry when the ice sublimates.**

We appreciate the reviewer's observation that this ice phase deposition is potentially a reversible process, where the melting or sublimation of the present ice would release the contained carbonyls. We have thus amended the line for specificity as "removed from the upper atmosphere". We do mention a few such studies that investigate deposition carbonyls. Specifically, we cite one study that investigates wet deposition carbonyls (Mu & Xu, 2009) and two that investigate deposition via snowfall (Bartels-Rausch et al., 2014; Dominé & Thibert, 1996a).

5. **Line 75. I would argue that 10 ppbv of a carbonyl is two orders of magnitude larger than mixing ratios in the middle to upper troposphere. For example, formaldehyde is typically 50 pptv in the background upper troposphere.**

We have amended this line as "approximately 1–2 orders of magnitude".

6. **Lines 330... I was wondering why formaldehyde has the highest ice partitioning coefficients compared to other compounds. What is it about each of the compounds that make it more or less likely to be taken up by ice. Since this topic gets addressed later (section 3.5), I suggest adding a comment here that an explanation is given later.**

We have added a line mentioning that this is discussed more thoroughly in section 3.5.

> Formaldehyde has the highest ice-partitioning coefficients in this study; the potential reasons for this are discussed further in section 3.5.

7. **Section 3.2. How do the results presented in this paper compare to previous literature (e.g., Winkler et al., 2002 that's cited in the paper)?**

The referenced citation for Winkler et al. (2002) (and the others in that line) is to mention surface adsorption as an insignificant or reversible process. They do not measure volume uptake. We do however compare to the measurements for aromatic hydrocarbons studied by Fries et al. (2007) in section 4.

8. **Line 472 mentions that a few factors were examined but only molar mass and heat of vaporization are discussed in the section. Could the results for the other factors be displayed in the supplement?**

We have included the rough plots for lnK vs lnPvap and lnK vs van der Waals volume in the supplement as Figures S5 and S6. These were not originally included as they are values mostly derived from MM and then can be considered collinear factors. The plots of lnK vs HPLC retention time and Henry solubility are both $r^2 < 0.3$ and are not suitable for publication due to their low correlation, but we have included them in the supplement as Figures S7 and S8.

9. **Lines 496-499 are interesting and really helps to explain the process. Are there references that should be cited that have examined the incorporation of trace gases into the ice crystal lattice? If so, please cite them.**

We have added a reference from Perrier et al., 2002, as it also explores the questions of whether a compound is incorporated into the ice crystal volume or phase separated to grain boundaries. There are a handful of studies that investigate trace acidic gases and their uptake (Dominé & Thibert, 1996b; Kärcher & Basko, 2004; Q. Wang et al., 2024), but do not necessarily explore their incorporation into the crystal. There is also some unpublished results from a group at Montana State University that was working with Prof. Mischa Bonn from the Max Planck Institute for Polymer Research, Germany on some crystallography studies of equilibrium growth ice impurities with simple acids like acetic acid.

10. **Lines 529-530. It states that one should divide the uptake coefficients presented in this paper by *RT* in order to get a coefficient that can be compared to other coefficients like the Henry's Law coefficient. I suggest that the authors provide that information in the supplement. Indeed, it would be useful to have**

**temperature-dependent coefficients listed so that it can easily be used by those exploring chemistry in/near ice clouds with a model.**

We have included these in the supplement.

11. **Line 530... How do the uptake coefficients compare to the $HNO_3$ uptake? The $HNO_3$ uptake onto ice is a well known process, so such a comparison will give readers a number to relate to.**

We have included a brief comparison to the uptake coefficients for HNO3.

Compared to the well-studied uptake of $HNO_3$, the uptake of most species studied here are 2-3 orders of magnitude lower while acetaldehyde, acetone, formaldehyde, and propionaldehyde may be considered in comparable range (Von Kuhlmann & Lawrence, 2006; Zondlo et al., 1997).

**Organization, Clarity, Technical Comments**

1. **I would like to see better construction of paragraphs. That is, each paragraph should start with the main idea (a topic sentence), followed by supporting sentences providing evidence for that idea and a concluding sentence to summarize or transition to the next sentence.**

We appreciate the reviewer's request for writing organization. We have reviewed the manuscript and adjusted where necessary.

**As an example, the paragraph on lines 40-48 begins with introducing carbonyls, giving a general explanation of their relevance to atmospheric chemistry. However, in the middle of the paragraph the topic changes to the lack of knowledge about removal of carbonyls from the atmosphere. Here, I would suggest splitting these two topics into two paragraphs and providing more detail on how carbonyls contribute to ozone and aerosol formation for their relevance, while the removal by deposition could be generalized to mention dry deposition, wet deposition via cloud water uptake, and ice deposition.**

We have made this adjustment along with our adjustments in comment #2.

**Please review the entire manuscript on paragraph construction as there are other places (e.g., lines 349-360) that have the same issue.**

We have reviewed these two paragraphs and reorganized their structure for clarity.

Every measured species except for MVK displays a strong correlation of ln(K) with inverse temperature and can likely be exclusively described by thermodynamic sorption. Furthermore, the uptake of these species can almost exclusively to attributed to codeposition during crystal growth as sorption on nongrowing crystals has been demonstrated to be insignificant or completely reversible for almost all chemical species studied. This specifically includes acetone, acetaldehyde, formaldehyde, and benzaldehyde (Fries et al., 2006; Hudson et al., 2002; Roth et al., 2004; Winkler et al., 2002). This observation of strong correlations with inverse temperature could indicate that K is controlled by transport, specifically if analyte transport is limited by accommodation at the ice-air interface (Davidovits et al., 2006; Jayne et al., 1991).

MVK however shows a weak negative trend with inverse temperature with a nonsignificant correlation. The absence of this correlation for MVK complicates this view of K controlled by transport. However, this could be explained by kinetic control resulting from transport phenomena occurring in either the gas or solid phases, i.e. processes that change the rates of transport of MVK relative to water rather than a K that is controlled by an equilibrium established between MVK and water. However, without sufficient evidence for a mechanism of kinetically controlled transport, the measurements of K here will be interpreted using equilibrium thermodynamics.

2. **There are a lot of acronyms. Be sure each acronym is written out the first time it is written (DNPH is not). In addition, be sure the acronym is needed, as it is easy to slip into the jargon of this specific topic and become less relevant for the broader atmospheric chemistry community.**

We have reviewed the manuscript for undefined acronyms. DNPH is defined at the beginning of section 2.3. However we have fixed this for SOA in section 1.

3. **Lines 121-125. It seems to me that these sentences about the PTFE tubing belong with "The second stage" paragraph (lines 103-116), which is where I was wondering about wall loss within the flow tube.**

PTFE tubing is not exclusive to the Flowtube section of the experimental setup. There is more tubing used in the gas mixing and gas measurement stages of the setup than there is within the Flowtube, so it doesn't seem accurate to move that line. Since measurements are taken as an integral of a full 24-48 hr experiment, wall loss is assumed to reach saturation early in the experiment.

4. **Line 192. Why not write ACN as CH₃CN?**

Since we are not discussing ACN's structure but its use as a solvent, it's preferable to use ACN or MeCN as shorthand.

5. **Tables: Why does the order of species listed start with MVK and then is alphabetical?**

That was done to single MVK out as the only outlying species that didn't follow the inverse temperature trends.

6. **Line 351. It should be, "exclusively be attributed to".**

We chose the phrase "described by" to reduce its repeat usage since we use "attributed to" in the following sentence.

7. **Line 351. What is co-depositing? Is it the individual carbonyl and water vapor, or is it multiple carbonyls depositing together?**

This is referring to the individual carbonyl and water vapor. For clarity we have added "codeposition with water vapor" to this line.

8. **Line 371. Please remind the reader why the thermodynamics results are being investigated. Try to write to the message you want the reader to learn from this section (i.e., starting with a description of Table 2 and 3 does not draw the reader into the science).**

We have added the following line as brief introductory sentence:

> The Van't Hoff analysis of uptake can be used to estimate the enthalpy and entropy of thermodynamic sorption.

9. **Lines 373-377. It seems like Table 2 should be discussed before Table 3.**

We have swapped the order of these lines accordingly.

10. **Line 412. Should it be, "which is attributed to weakened ..."?**

This has been fixed in the manuscript.

11. **Line 417. Is there photodegradation in the flow tube?**

The Flowtube is housed inside an insulated chamber with very little likelihood of photodegradation. The gas mixing stage is however in a typical laboratory environment which has the potential for photodegradation.

12. **In the supplement, what is IKR?**

Isokinetic relation; this acronym is defined in the first paragraph.

**Figures and Tables**

1. **Table 1. Please remove the banded background coloring as it makes it harder to read. It would be nice to have a little more space between rows. Another thing to think about is to present all values greater than 1 without the exponential part (e.g., HCHO at -40degC written as 103 +/- 63. That way the discussion (line 332) about net uptake is easy to see in the table.**

The banded format for the tables is a personal preference. If accepted, the tables will likely be reformatted for the publisher's styling choices.
We also originally had the values greater than 1 formatted as the reviewer suggests; but I was strongly encouraged by my coauthors to maintain the current formatting for consistency.

**RC2:**

**The authors describe a laboratory setup and use it to characterize the amount of organic vapor incorporated into ice grown by deposition. This reviewer participated in a similar study ~ 20 years ago. These are not easy measurements to make. Interpreting the results is also difficult.**

**One of my major critiques concerns the tentative conclusion that a surface layer is the "single dominant influence" on uptake. This runs counter to the discussion of how uptake might be controlled by accommodation or affinity within bulk ice.**

**The authors should consider my critiques and reply with a revised manuscript.**

We greatly appreciate Dr. Snider's review. We find their insight to be invaluable and critical to this review process. Their critiques have been taken with thorough consideration and applied to the revised manuscript.

We would like to first address that any insinuation that a surface layer is the "single dominant influence" on uptake is unintentional. We think two separate ideas concerning entropy-enthalpy compensation (EEC) may have been conflated to give a false impression of what EEC may tell us about the uptake process. There are still many conflicting ideas in the literature on what EEC may represent, and as a result discussing EEC in-depth can be generally controversial and often confusing. Our discussion on EEC in section 3.4 is central to understanding what this phenomenon could mean for the uptake process but ultimately cannot be comprehensive to understanding EEC in its entirety nor its controversies. We have tried to mitigate any confusion by rewording some instances where it may be unintentionally implied that a surface layer is the single dominant influence on uptake. However, we do recommend exploring some of the literature on EEC that is referenced in the manuscript in order to better understand what EEC could mean in this context.

Referenced literature on EEC: (Grunwald & Steel, 1995; Krug et al., 1976; Leffler, 1955; Leung et al., 2008; L. Liu & Guo, 2001; Moulik et al., 2019; Pan et al., 2015; Sharp, 2001).

**Major Critiques:**

**On L276, you say that "Positive values of DG indicate the analyte favors the gas phase while negative values of DG indicate the analyte favors the ice phase. Even lower, more negative values of DG would indicate more efficient uptake of the analyte into the ice phase." Also, on L330, you say "Values of K > 1....indicate net uptake of the**

compound into the ice phase. Conversely, values of K<1...indicate negligible uptake and that the compound favors remaining in the gas phase." And, on L376, you say that "...DG>0 indicates unfavorable uptake of the species into the ice phase while DG<0 indicates favorable uptake in the ice phase."

As I demonstrate below, your statements are illogical for a system with an amount of ice which is magnitudes smaller than the amount of gas. I will symbolize these amounts as volumes (Vs and Vg) and will insist that Vg/Vs >> 1. Also, I note that Vg>>Vs is the situation within clouds. With these constraints, and assuming sufficient time for equilibration, the K you report allows for determination of analyte amount (as a mole count) within the solid (KCgVs), within the gas (CgVg), and the fraction of analyte within the solid.

Fraction of analyte within the solid = KCgVs / (KCgVs+ CgVg) = K / (K + Vg/Vs)

My point is this: Since K is not large compared to 1 (Fig. 4), and Vg/Vs >> 1 is the situation within clouds, it does not make sense to say that "negative values of DG indicate the analyte favors the ice phase", or that "Values of K > 1....indicate net uptake of the compound into the ice phase", or that "DG<0 indicates favorable uptake in the ice phase." Also, while it is logical to say that "Positive values of DG indicate the analyte favors the gas phase", and that "Conversely, values of K<1...indicate negligible uptake and that the compound favors remaining in the gas phase", and "DG>0 indicates unfavorable uptake of the species into the ice phase", these last statements are also true for a K that is magnitudes larger than 1.

In summary, you need to rewrite these sections of text. A suggestion: Think of DG is a placeholder for K, and not as an indicator of how uptake alters the partitioning of the analyte within clouds. If you do not want to focus on clouds please recognize that it is an unusual experimental setup that allows for Vg/Vs ~ 1 while also assuring time for equilibration.

We greatly appreciate Dr. Snider's thorough evaluation of how we applied partitioning for its relevance to net uptake. The mass imbalance concerning water content within clouds was neglected and is critical to how partitioning influences uptake. We have rewritten these sections of text to reflect this. These statements were originally intended to explain the relevance of K and dG for readers unfamiliar with thermodynamic equilibrium but was oversimplified in the process. While we have amended this section to not overstate uptake's influence on net transport, we affirm that uptake on ice may still be a relevant influence on vertical transport. We have reviewed Brimblecombe and Dawson (1984) as recommended, notably sections 3,4, and 5. While Vg/Vs >> 1 under the typical conditions

of cloud water content, a ln(K) > 10 ($10^3$ mol $l^{-1}$ atm$^{-1}$) still represents a non-negligible analyte fraction in the deposition phase. While true that we don't observe any partitioning coefficients that high in this temperature range, extrapolating our data to colder cirrus cloud temperatures such as to -60 C or even -80 C, species like formaldehyde, acetone, or acetaldehyde could potentially reach a value of K necessary for significant transport. We have included a brief discussion of this in the conclusions as a limitation on the application of our conclusions:

> Importantly, the partitioning of these compounds to the ice phase are governed in part by the volume ratio of the partitioning phases and may be less relevant in situations where the amount of ice is magnitudes smaller than the amount of gas, which describes nearly all natural cloud conditions (Brimblecombe and Dawson, 1984). Specifically, to appreciably accumulate a significant fraction of a compound from the gas phase, a large K is needed in order to overcome the difference in ice and air, likely a ln(K) > 10

We have rewritten these sections of 2.6 and 3.2 as follows:

> These values describe the thermodynamic potential of the uptake process. Positive values of ΔG indicate the analyte proceeds spontaneously to the gas phase while negative values of ΔG indicate the analyte proceeds spontaneously to the ice phase provided similar magnitudes of ice and gas volumes. Even lower, more negative values of ΔG would indicate more efficient uptake of the analyte into the ice phase provided the available ice volume is sufficient.

> Values of $K > 1$ or ln($K$) > 0 indicate net uptake of the compound into the ice phase for a system with equal volumes of gas and ice. Conversely, values of $K < 1$ or ln($K$) < 0 indicate negligible uptake and that the compound favors remaining in the gas phase for a system with equal volumes of gas and ice.

> In Table 2, a Δ$G$ > 0 indicates unfavorable uptake of the species into the ice phase while Δ$G$ < 0 indicates favorable uptake into the ice phase for a system with equal volumes of gas and ice.

**Related to the previous comments and critique:**

1) **Since partitioning is strongly on the side of gas-phase, I don't accept your assertion that uptake into ice "...may be relevant as an influence on vertical tracer transport" in the Conclusion.**

We have rewritten this statement to be more specific to the circumstances in which ice uptake may be relevant.

2) **My formulation of partitioning comes from Brimblecombe and Dawson (https://doi.org/10.1007/BF00127265). As far as I can tell, they were the first to put the concept into the literature.**

3) **There is a treatment of partitioning in the atmospheric chemistry textbooks. One of these is Lamb and Verlinda (Physics and Chemistry of Clouds, Cambridge University Press, 2011; see pp. 166-168).**

We thank Dr. Snider for their recommendation of relevant literature. We have reviewed both texts and applied their conclusions to our revisions.

4) **I recommend that you report your ice uptake coefficient with dimension of mol m-3 Pa-1. This is your framework for the Henry Law discussion (Equations 5 and 6). With this change you can eliminate a confusing sentence (L210-L211). It would also eliminate the need for a translation from a dimensionless K to dimensional K, in the Conclusion.**

We have added a table of the uptake coefficients in units of mol $m^{-3}$ $Pa^{-1}$ to the supplementary materials. We reported unitless coefficients as a more direct comparison to Fries et al. (2007) and for analytical simplicity, which we will maintain in the manuscript. Line 210 is observing that this analytical simplicity circumvents the need to reference external standards for true quantitation. We have rewritten this line to reduce confusion.

> Since the ice-gas partitioning coefficients are unitless and the sample matrices are the same, only relative quantitation is necessary for the calculation. This circumvents the need for true quantitation as referenced to an external standard.

**L134 says that Si = 1.5 was a constraint in all experiments. This implies that either deposition or condensation can occur. I checked my assertion here:**

| T,[K] | Si,[-] | ei(T),[Pa] | e,[Pa] | es(T),[Pa] | RH,[%] |
|---|---|---|---|---|---|
| 253.15 | 1.50 | 103. | 155. | 125. | 123. |
| 243.15 | 1.50 | 38. | 57. | 51. | 112. |
| 233.15 | 1.50 | 13. | 19. | 19. | 101. |

**Note that the relative humidity (relative to saturation over liquid water) exceeds 100 % at all temperatures (-20, -30, and -40 oC). Since you are using "tank" air, it is unlikely that droplets formed, impacted, and contributed to the ice deposit via riming.**

**I recommend that you consider the following as a process that may have occurred in your experimental setup:**

**Provided some regions on the glass are clean (i.e., no ice nucleating particles) you may be condensing liquid directly from the vapor. Freezing would then occur, for example, once a liquid domain touches an ice domain. The latter could have been previously frozen or previously deposited. In either case, the existence of liquid water would rationalize your exothermic uptake coefficients. Here, I'm assuming that Henry's law uptake (into the liquid) increases with decreasing temperature below 273 K.**

We have included this as a potential limitation in the conclusions. We agree however that it is unlikely that liquid droplets formed and impacted via riming. We took precautions to prevent droplets formed in the gas mixing stage from being introduced into the Flowtube, but liquid droplets that condense once inside the Flowtube are harder to prevent. We had considered this at one point in the experimental design and planned to sample only certain sections of the Flowtube to mitigate this potential issue. This way samples that contained more condensation formed droplets, which we believed to be likely closer to the inlet, could be compared to the samples collected closer to the outflow. This could also then be used to compare along the temperature gradient along the Flowtube. However, since ice yields at these saturations are so low, samples had to be collected from the entire length of the Flowtube in order to collect enough mass for analysis.

Also, if liquid water condensation was a major contributor to these measurements, we'd expect Henry solubilities to be the dominate factor controlling uptake and that more soluble species like glyoxal to have much higher uptake. We have added this discussion to the conclusions:

> These measurements are exclusively a description of the gas to ice solid solution equilibrium and neglect investigation of gas to liquid or liquid to ice equilibrium. Critically, the measurements here could be influenced by liquid water condensation within the Flowtube. Provided some regions of the glass substrate are clean and free from ice nucleating sites, some liquid phase water could condense directly to the substrate and then freeze thus competing with the diffusional growth ice. Additionally, some liquid water could condense into droplets within the gas stream and then impact on the substrate thereby contributing to the deposited ice via riming. These alternate modes of freezing are unlikely to occur under these conditions as the clean carrier/dilution gas has virtually no nucleating particles for liquid droplet condensation and the glass Flowtube substrate is ideal for facilitating dendritic ice crystal growth (Chen et al., 2020). However, the presence of liquid water condensation within the Flowtube could be an alternative explanation to the observed exothermic uptake trends, assuming that Henry's law uptake into the liquid phase increases with decreasing temperature below 273 K. If this were the case, Henry solubilities should be the dominate factor controlling uptake and it

would also be expected that more water-soluble species like glyoxal would have much higher uptake coefficients than less water-soluble species like formaldehyde. Neither of these are observed here so it can be concluded that liquid water condensation within the Flowtube is unlikely to have significantly influenced the present measurements.

**Related Recommendations:**

1) **I recommend that you refer to your measurements as an uptake coefficients, as sorption coefficients, or as an ice-gas partitioning coefficients. You complicate the reading by using all of these. The place to do this is in the Introduction not in Section 2.6.**

We have reviewed the manuscript for inconsistent uses of the three terms. The following comment concerning the term "partitioning" its competing usage to describe analyte mass continuity has moved us to use a different term. We have replaced the instances of its use with "uptake coefficients" or "volume uptake". "Sorption coefficient" is only used twice and in specific reference to the thermodynamic sorption calculation. We have edited these lines to be more clear that the uptake coefficient is standing in for a strict "sorption coefficient".

Practically speaking, the uptake coefficient $K$ can also be used as a sorption coefficient or a dimensionless uptake coefficient with respect to the removal of trace gases in the upper atmosphere.

Continuing the thermodynamic analysis, the theoretical temperature dependence of the uptake coefficient $K$ —used in place of a sorption coefficient—can be determined with the van't Hoff equation, which when substituting with the Gibbs-Helmholtz equation produces the following:

2) **Since the cloud chemistry community uses "partitioning" to describe analyte mass continuity, within an element of cloud, I recommend that you not use partitioning to modify the coefficient you are evaluating. Similarly, I encourage that you do not use "partitioning" as a place holder for the process you are investigating.**

Shared terms with different definitions between meteorology and chemistry is an issue we've encounter already with terms like "precipitation" and "deposition". We do our best to try to avoid conflating such terms but it's unavoidable in certain cases. While the term

"partitioning" is accurate and descriptive from a chemical standpoint, we have swapped the use of "gas-ice partitioning" for "volume uptake" throughout the manuscript to avoid its confusion with other uses of the term.

**3) I do not think you need Equation 2. There are tables of saturation pressure in the textbooks. E.g., Rogers and Yau, Third Edition, Elsevier, 1989; Table 2.1. The important thing is that you check your Equation 2 for dimensional consistency and for adequate numerical precision compared to tabulated data.**

We have moved Equation 2 to the supplementary.

**4) There is no reason to report the gas constant with six-digit precision or to report the reciprocal temperatures (p. 14) with four-digit precision.**

While such precision is unnecessary, we did want to report our calculations rigorously and thus we only round to significant digits after the calculation.

**5) The density of ice can be taken to be a constant.**

Yes, the difference in density is less than 0.2% and could be rounded 0.92 g cm$^{-3}$. However we did want to keep our calculation thorough, and rounded to significant digits afterwards.

**Your discussion on L175-L187 should be revised. You are using aqueous solutions to generate a prescribed amount of gas-phase analyte within the flowtube. The [X]aq (Equation 6) is what's required to produce a specified gas-phase mixing ratio (10 ppbv). So, I recommend that you put the target gas-phase mole fraction (10.x10-9), and your pressure, into Equation 6.**

We have adjusted the line preceding Equation 6 to make this clearer. Since experiments are performed at atmospheric pressure, the mixing ratio and mole fraction are equivalent we feel that the $p_x$ accounts for the target gas-phase mole fraction and pressure.

**Please note:**

**L179. You are producing a partial pressure, within the flowtube, not a "gas concentration." This should be corrected here and elsewhere.**

We have corrected instances of 'gas concentration' to partial pressure throughout the text.

**It's not clear why you have the superscript "cp" in Equation 6. Also, it's quite common to use Kh, not H, for the Henry solubility, at least in the cloud chemistry community.**

It's just the concentration-pressure defined H. We used H as it is in line with R. Sanders 2023 and keeps K more distinct for the uptake coefficient.

**About "breakthrough." I don't understand your statement that Kahnt et al. (2011) observed larger breakthrough, at lower relative humidities, but your absolute humidity is much lower. Do you mean that Kahnt et al. (2011) observed larger breakthrough at _low_ relative humidities? Additionally, relative humidity and absolute humidity are related, but it is not clear how to compare your humidity condition (absolute) to the humidity condition in Kahnt et al. (2011) (relative).**

We have revised this phrase to be clearer. Our absolute humidities are lower than Kahnt et al. (2011) while our relative humidities are higher. They are measuring larger breakthrough potentials than us. They observed more breakthrough at higher relative humidity than at low relative humidity. We are trying to concisely explain that our lower breakthrough potential at a higher relative humidity is because our absolute humidity is much lower than theirs. We have rewritten the line as:

> While Kahnt et al. (2011) observed much higher breakthrough potentials than this at lower relative humidities than in these experiments, the absolute humidity in these experiments is lower by 5 orders of magnitude. Since water can both encourage and inhibit the derivation, any changes in humidity conditions may alter the breakthrough potential of any of the analytes.

**Please note the title of Kahnt et al. (2011): "Denuder sampling techniques for the determination of...derivatization methods." There are places where you use the modifier "derivation" instead of "derivatization." Please check throughout.**

We have fixed instances of "derivation" and replaced them with 'derivatization' throughout the text.

**The word "massed" is used. Do you mean weighed? It seems that the mass of the ice was derived by weighing together with additional information. On L225-233 you say that the flow tube was rinsed (methanol), that the extract was "massed", and that the solution was evaluated in a refractometer. The acquired information is sufficient for determining the mass of H2O that was extracted from the tube.**

**1) Please correct/change all instances of "massed".**

We have fixed instances of "massed" in favor of "weighed" where appropriate throughout the text.

**2) You refer to the "collected ice mass" on L159. Should this be "sampled ice mass" or "extracted sample mass"? There are other instances of "ice mass". You should consider changing these, for clarity.**

We have swapped instances of "collected ice mass" to "sampled ice mass". There is only one instance of 'ice mass' and it specifically refers to the deposited ice mass and not the sampled ice, so we deem it appropriately used.

**3) On L305-L309, you report the calculated ice mass (Equation 3) and you compare to the collected ice mass (aka, the "yield."). In your procedure, it seems, the methanol does not capture all the ice. Hence, the expectation is calculated > measured. Looking at data from the -30 and -40 oC, that expectation is verified. In our paper (Huffman and Snider 2004), a calculation overpredicts the observed ice mass. Splintering of the ice sample, during the uptake experiments, was suspected. In your experiments, do you have evidence of splintering?**

We reason that the majority of uncollected ice is deposited on non-extractable surfaces rather than splintering. We think there is simply more available surface area for deposition than large deposits that can splinter. If splintering occurred, then it should reason that with experiments more deposition with larger deposits should have more splintering and thereby a larger difference in theoretical/actual yield (specifically larger theoretical than actual). However there is a larger difference at -40 with the lowest deposited mass than at -30 and at -20 the actual is larger than the theoretical. So splintering is not a particularly good explanation for the differences in measured and calculated mass.

**4) Your visual observations (L318-319) are contradictory. The coatings were "either over the entire surface" or "not evenly distributed." Which of these best represents what you saw?**

Something can be thinly coated over a surface and still not be evenly distributed over that surface. These are not contradictory descriptions. We did take out the term 'entire' as it might unduly imply uniformity.

**5) All one can visualize is a macroscopic ice deposit, not the (microscopic) nucleation sites. Accordingly, I recommend that you revise L321.**

We have swapped the term 'nucleation sites' with 'ice deposits'.

**Contamination during the processing of the samples (the H2O samples and the cartridge samples) is a possibility. You addressed this on L325-L328. Your approach is to process blank ice samples grown from liquid.**

**1) Why not collect/process "blank" ice samples taken from the flowtube when it's operated with no analyte?**

This was performed. The lines in 325-328 describe this. We have added the phrase "grown in the flowtube' to eliminate confusion.

**2) You say that the blank signals were in the same range as analytical blanks, which were all below detection limit. It's my opinion that you should tabulate the analytical blank values - or tabulate their average and standard deviation – and tabulate the same for the blanks ice samples.**

While these values would be useful to see from an analytical standpoint, they don't carry information that is useful to conveying the message of the text. We can include this with the raw data that is provided on request.

**I may have missed this. Did you do an analysis of analyte amount in denuder #1, denuder #2, and in the flowtube and use that information to quantify error in K coming exclusively from the chemical analysis method? This seems like a useful thing to do....mass must be conserved.**

We did this analysis in our early experiments but it was not included in the publication. It wasn't particularly rigorous and the information it provided is essentially the same as applying the breakthrough potential to the analytical uncertainty. The difference between partial pressures measured from denuder #1 and #2 weren't significantly different, and so their measurements were averaged to provide the average partial pressure along the flowtube. The point being, the deposited sample mass in the ice phase is much smaller than the large mass collected in the denuders such that no significant difference can be seen between the input and exhaust from a mass balance perspective.

**L39 - Neither Gautam et al. (2025) Seymore et al. (2025) are in the bibliography.**

This has been fixed. These papers were in preprint at time of submission.

**L502 – Neither Gautam et al. (2024) Seymore et al. (2024) are in the bibliography.**

These are the same papers. These papers were in preprint at time of submission.

**Regarding your tentative conclusion that a surface layer is the "single dominant influence" on uptake. This was examined by Valdez et al. (https://doi.org/10.1029/JD094iD01p0109). This reference is not in your bibliography.**

We have added this reference to the text in line 491.

> ... previous uptake studies also use aqueous film models to account for $SO_2$ capture (Valdez et al., 1989).

**Less-than-major Corrections and Less-than-major Critiques:**

**"ACN", define where this acronym is first used.**

This is defined at the beginning of 2.3 Chemicals and Materials, line 204.

**L144. Remove "using the hygrometer."**

Removed.

**L150. Remove "which is equivalent to the diffusive flux."**

Removed.

**L153. A cloud physics textbook (Rogers and Yau; Third Edition, Elsevier, 1989) says that the diffusivity is 20% smaller at -40 oC compared to -20 oC.**

We did not account for changes in diffusivity in our estimation of ice growth rate. A back of the envelope calculation says the difference in J from a 20% difference in Dt is roughly 10%.

**L157. Please see my earlier comment about condensation and deposition. You are interpreting tk as the time for "deposition", correct?**

Correct. This is also done by Fries et al. 2007

**L202-L204. Is "hydrazone" different from "hydrazine" in DNPH?**

Yes, DNPH is a hydrazine, a compound with a nitrogen-nitrogen single bond. Once reacted with a carbonyl, it forms a hydrazone, a carbon-nitrogen double bond adjacent to a nitrogen-nitrogen single bond. The hydrazone is what is measured in these experiments.

**L262-L267. I recommend that you not use "Cice" for the absolute mass of analyte in ice. Also, is the modifier "absolute" implying something? Recommend that you remove "absolute."**

This is also the convention used by Fries et al. (2007). $C_{ice}$ is likened to concentration and there is already an 'm' used by $m_{ice}$. I suppose $m_{x,ice}$ could be used or maybe $X_{ice}$ but I think $C_{ice}$ is fine. 'Absolute' is being used to state 'total' and not a relative concentration.

**L275 You say: "These values describe the energy available for the uptake process." I do not agree with this. Think of DG is a placeholder for K. Also see my first critique.**

We have rewritten this line to avoid a misleading simplification.

> These values describe the thermodynamic potential of the uptake process.

**L361 "...and water". Do you mean "...and ice"?**

Yes, we have edited this line as follows:

> i.e. processes that change the rates of transport of MVK relative to water vapor rather than a K that is controlled by an equilibrium established between MVK and ice.

**L411 "Endothermically" or "nonexothermically"? Be consistent.**

These are terms with different meanings. We think the use of 'endothermically' in line 411 and the use of 'nonexothermic' in line 391 are both appropriate.

**L416-L417 I don't understand how photodegradation of MVK, within the atmosphere, can be the cause of the weak correlation with inverse temperature, in your laboratory study.**

This discussion highlights MVK's reactivity and that its photodegradation within the bubbler over the duration of the experiment may influence the measurement.

**L430 "artifactully". Is there a better way to say this?**

We have swapped "artifactully" for the phrase "as a statistical artefact".

**L435 The letter "H" is being used to represent several properties. There is the Henry solubility (Equation 5), the enthalpy of solution (BTW, some refer to this as the enthalpy of gas-to-liquid transfer), and the enthalpy 158 change you derive via the van't Hoff regression. You can avoid introducing another "H." Do that by acknowledging that the enthalpy change you derive for a rather cold experimental temperature may be significantly different from the reference-state enthalpy. Related to this, why not represent the Henry Solubility (Equation 5) with Kh? That is common, and it would eliminate one of the "H" symbols.**

We have rewritten this line to clarify the idea without having to introduce another H term. As also answered in a previous response, we have opted not to represent Henry Solubility with Kh as we use K for the uptake coefficient, which is a central focus of the manuscript. Essentially, we'd rather have a reader be confused on the specifics of what an H term represents rather than confused on the specifics of what a K term represents. We have rewritten the line as follows:

> Additionally, van't Hoff analysis assumes that the enthalpy change relative to the reference-state enthalpy change is negligible (i.e., the heat capacity change is negligible). This may be invalid for an experimental temperature which may be significantly different from the reference-state enthalpy (Leung et al., 2008).

**L480 Which "solubility"? I think you are referring to the Henry Law solubility, but I'm not sure.**

We were referring more specifically to aqueous solubility but that's a term included in the Henry Solubility. Have added 'aqueous solubility' to clarify here.

**L456 Is this the first occurrence of "additivity"? It is not clear what you are implying by "additivity"**

"Additivity" is a specific term to the modeling of thermochemical properties of molecules. Thermodynamic additivity is the principle that if two components, A and B, contribute independently to some process, then the total change in free energy (or enthalpy or entropy) is the sum of components, $\Delta G = \Delta G_a + \Delta G_b$. This description is from Dill (1997). We have clarified this line as:

> ...that there is a single source of additivity for the series of compounds studied (Lumry, 1995); i.e. a single thermodynamic component that controls the uptake process.

**L457 I recommend something like this: "...could help to generalize descriptions of the uptake process in models"**

We have incorporated this recommendation into the manuscript.

> ...it seems possible that the EEC seen here could be used to help generalize descriptions of the uptake process in models.

**L539 It is not clear what you are implying by "..void space."**

It is the unoccupied space in the crystal volume.

**L711 Check the format of the Warhaft reference.**

We have fixed the formatting of the Warhaft reference.

**RC3:**

**Summary**

**This article discusses a study aimed at measuring gas-ice partitioning coefficients of 14 carbonyl gaseous species onto crystalline ice surfaces. Through compositional analysis of the ice crystals after gaseous carbonyl exposure, insights into uptake behavior are gained. These insights are gained via: 1. the catalysis of ice crystal growth on the walls of a flow tube apparatus at temperatures that mimic those in the troposphere (-20, -30, and -40 C̊), 2. vapor pressure dependence studies of each carbonyl species and how it plays a key role in diffusional uptake onto the ice crystal, 3. the determination of partitioning coefficients and using Van't Hoff analyses to calculate the entropy and enthalpy of uptake, and 4. providing intercomparison studies produced by other researchers to help draw conclusions and provide reasoning to their overall findings and shine light on knowledge gaps that need to be further explored. The authors provide data covering various factors, such as temperature and vapor pressure, and how these factors contribute to the diffusional deposition of gaseous carbonyls onto ice surfaces. Although an in-depth study was reported, I had some questions that I feel should be addressed before the article is published.**

**Comments:**

1. **Line 67: replace $K_{g,ss}$ with $K_{l,ss}$**

   This has been fixed in the manuscript.

2. **Were the gaseous species cooled to experimental temperatures before entering the flow tube? If flowing room temperature/warmer analytes into a chilled flow tube, wouldn't it take time for the analyte to reach lower temperatures, skewing the vapor pressures (especially since the temperature in the flow tube varies at certain distances from the inlet)? The gas analyte would exist at the same temperature as the ice crystals in the atmosphere. Is the time it takes to chill the gaseous analyte accounted for in the flow tube? What is the residence time of the analyte within the flow tube?**

The gas stream was not prechilled before introducing into the flowtube. The gas temperature does vary along the flowtube. This thermal gradient is presented in the supplement in Figure S1. It is important to note that the thermal mass of the flowtube is much larger than the gas stream, so at the interface of the gas and substrate it is

reasonable to assume that the gas temperature is at the experimental temperature. The interior volume of the flowtube is 1L and the flowrate through the flowtube was typically 4 lpm, so the residence time is roughly 15sec. The relevant vapor pressure also should not vary as we perform our measurements as volumetric mixing ratios.

3. **Line 74: If I am reading this correctly, it seems that uptake experiments done using mixtures of carbonyl compounds, rather than individual components. How can the authors be sure that there are no cooperative or competitive effects impacting the uptake equilibria? That is, some compounds might adsorb more preferentially and block an adsorption site or displace another carbonyl that may bind more weakly to the ice. Likewise, could it be that some species bind forming a monolayer of organics that and now create a more favorable surface for subsequent VOCs to bind to. Have the authors done the requisite experiments to investigate how the uptake depends on concentration? I would expect that uptake would decrease as the concentration of VOC increases, so total concentration of the mixture or of the individual components will be important. Ideally, one would work with one organic component at a time and at low concentration so it can be assumed that interactions between a specific VOC and water in the ice are the only interactions that need to be considered.**

Yes, these experiments were performed as mixtures of carbonyls. This experiment was first performed with a single component solution of benzaldehyde but at different water vapor saturations and a higher partial pressure. This data is not published as it's not directly comparable to the data presented here. Huffman and Snider (2004) saw no evidence that single component uptake is any different than mixtures and our measurements for benzaldehyde are not significantly different either. Total organic concentration could be a significant factor in altering uptake, however we don't have any evidence of uptake inhibition at these concentrations and natural ambient concentrations are even lower than what we replicate here.

It is important to note that each measurement operates on the order of days to ensure adequate ice crystal growth in addition to the sample processing and analysis. A single experiment may then take up to a full week, assuming everything is operating correctly. It's logistically unfeasible to run so many single component measurements at the necessary temperature ranges with enough replicates for statistical validity.

4. **Line 225: Was there gas analyte trapped in the flow tube at the conclusion of the experiment and during transfer to the cold chamber? Would you predict that**

**would alter the composition of the ice crystal if residual analyte was trapped for longer times?**

It's possible at the conclusion of the experiment, the parcel of gas in the flowtube may have a longer time to equilibrate with the substrate. This is on the order of minutes before sample extraction. Because the experiment operates on the order of days, this is a very small time for any remaining deposition to occur. Crystal growth is also assumed to be at equilibrium so excess time for equilibration also shouldn't influence the concentration of analyte present in the ice.

1. **Line 310: What are the % loss of carbonyls on the apparatus/non measurable surfaces? If these losses were measured, were they used to correct/account the concentrations measured?**

We assume that any analyte loss due to adsorption or wall losses reaches saturation very quickly relative to the experiment duration; i.e. saturation occurs within the first 15 minutes while the experiment lasts over 24 hours. Because these are integrative measurements, this analyte loss can be considered negligible.

2. **Line 343. Discussion of potential loss of formadehyde seems to fit better in the conclusion section.**

We have kept this line to maintain the relevant context for formaldehyde in this section. We have however expanded this discussion in the conclusion as:

> Notably, as these species are relevant as significant sources of OH radicals in the upper troposphere (Cooke et al., 2010; Fried et al., 2016), it then is likely that this ice uptake could be a significant influence on OH radical formation in the upper troposphere at temperatures less than the range studied here, notably $-60\,^{\circ}C$ where the extrapolated uptake could reach $ln(K) > 10$.

1. **Was optimizing flow tube surface area, length, etc.. studied to increase ice crystal growth uniformity?**

Unfortunately altering the substrate geometry is quite difficult. Any adjustments to the glass substrate requires a glassblowing technician and then this geometric has to be constrained to the way the tube can inface with the cooling coils. So testing such parameters would become costly and challenging. Since the growth conditions of the ice is not the central focus of this experiment, exploring the flowtube geometry's impact on ice crystal morphology was not studied. We did find that on average, the vapor deposition efficiency—that is the percent difference between the input and exhaust water vapor

concentration, presumed to be the percentage of water deposited as ice—was almost always 46%. This value never deviated more than 8% over the course of all experiments. This is potentially a geometric constraint of the flowtube apparatus as this value did not appear to change with temperature, flow rate, nor experiment time.

1. **Is there a better method for studying deposition that allows for better ice crystal uniformity and collection without use of many solvents and steps? Are you concerned with analyte loss during recollection methods?**

The major limitation on studying deposition under these conditions is the very low crystal growth rate that occurs at these water vapor saturations under low temperatures. This produces sample yields that are not viable for sampling methods like physical scraping. In 3.1, the crystal growth rates are on the order of mg per hour to produce total ice yields that are still a gram or less spread over a surface of 785cm^2. Even if you increase the water vapor deposition efficiency to 100%, you're only increasing the ice crystal growth rate by a factor of 2.

We collect our samples cold, i.e. in a cold chamber typically at -10C. The derivatization method we use also prevents the majority of analyte loss during recollection. The DNPH hydrazone is vapor stable, so once the reagent is added, analyte loss to the vapor phase should be negligible.

1. **Page 16, Figure 2: Individual data points are difficult to identify in the plot. Is there another way to show the data and increase ease of identifying each point?**

We considered changing the markers for compound initials but that looked even more cluttered. Without breaking the plot into several individual plots (which all mostly look the same and would be redundant except for MVK), it's difficult to declutter this plot. We can present the individual plots for each compound in the supplement if that would be helpful. However, the main point of Fig 2—that all species except MVK behave exothermally—is still fairly clear in this format.

1. **Page 19, Line 416: You mention photodegradation as a potential factor for weak correlation with MVK, but did you preform any studies with your system to further confirm or deny this? Do you plan to conduct photolysis studies with these systems in the future?**

There are several other that describe photodegradation of MVK (Kato & Yamazaki, 1976; Renard et al., 2014; Weerasinghe et al., 2024). To thoroughly study this process is outside the scope of this research. However, we may in the future take precautions to prevent

photodegradation by ensuring that the bubbler system and gas mixing stage of the experimental setup are insulated from stray UV that could influence the input partial pressures. We may also modify the setup to introduce photolysis in a controlled manner to investigate this potential factor.

1. **Line 484: Carbonyls can't H-bond between themselves, but they can interact strongly via dipole-dipole interactions. How does this figure in the discussion here?**

Keesom interactions could be present, but unlikely to be major contributors as solvent-solute interactions are dominant over solute-solute interactions at these concentrations. However, if these potential dipole-dipole interactions are stabilizing in the condensed phase, we should see an enhancement of ice phase uptake at higher concentrations. This would be less likely to occur in the natural atmosphere as ambient concentrations of these species are lower.

2. **Line 492-500: I have questions about this paragraph, where the authors are speculating on the mechanism that is leading to the observation that uptake tendency into the ice is inversely proportional to molecular size. My first question pertains to the sentence, "in order to be taken into the ice phase, a compound must be incorporated into the ice crystal lattice structure." A crystal structure is comprised of molecules packed into a particular lattice. Therefore, I believe the phrasing to use is just "the ice crystal structure." If that is true, then are the authors suggesting that the smaller ketones are cocrystallizing with the ice? Or are they suggesting that the molecules are deposited in grain boundaries and more adsorption occurs when defects are enhanced? It seems to me that co-crystallization is very unlikely for ketones + water as this would result in a completely new and unique crystal structure. I believe it is more likely that the ice crystals grown from vapor deposition are mesoporous with a high surface area and that the behavior the authors have demonstrated (increased tendency to partition to ice with decreasing molecular size) describes the process of adsorption into mesoporous materials, where larger molecules are sterically inhibited from entering the small pores, while small VOCs diffuse more easily leading to much higher adsorption capacities. If the ice surfaces were exposed to a cocktail of VOCs, would there would be a preference for the smaller VOCs adsorbing into the pores first, and the filled pores would then exclude other molecules from filling in.**

We are speculating that the carbonyls could be incorporated in the ice crystal volume, either in the crystal network or in the void space as defects. As far as we understand, ice crystal impurity point defects are not new or particularly unique and has been a potential explanation for other ice behaviors (Perrier et al., 2002) as well as studied for some solutes (Ballenegger et al., 2006; Cwiklik et al., 2009; Eichler et al., 2019). If deposition occurred entirely in grain boundaries, what would be the reason for the preference of smaller molecule uptake? This mesoporous explanation of the size preference in uptake appears viable. We have also included this explanation in the discussion in this section. If the reviewer has literature that would be helpful in expanding this potential explanation, we would be happy to receive it. Though, we are skeptical of the idea that the presence of small compounds inhibit the uptake of larger ones, as we have discussed previously on the influence of mixtures versus single component experiments. We have added this discussion to section 3.5:

> An alternative explanation could be that the preferential uptake of smaller compounds is due to possible mesoporous conditions on the ice surface; i.e. pores that develop on the ice surface prevent the accommodation of larger compounds which are sterically inhibited from entering small pores.

3. **Line 526-528: It is stated that the K-values measured in this study are below the Crutzen and Lawrence threshold for being substantial atmospheric removal processes. But what about formaldehyde, which according to line 343 has a high tendency to deposit in ice (see comment above, this may be a better place to place that discussion).**

We have expanded our discussion on formaldehyde here as follows:

> Notably, as these species are relevant as significant sources of OH radicals in the upper troposphere (Cooke et al., 2010; Fried et al., 2016), it then is likely that this ice uptake could be a significant influence on OH radical formation in the upper troposphere at temperatures less than the range studied here, notably $-60\,^{\circ}C$ where the extrapolated uptake could reach $\ln(K) > 10$.

---

## Author Response (AR2)

**1. Line 75. I suggest changing the sentence to 2-3 orders of magnitude. Sorry, but convective outflow and UT background mixing ratio measurements of the most abundant carbonyls do not show an order 1 comparison (1-10 ppbv) to the mixing ratio used in the lab experiments.**

This change has been made in the manuscript.

**2. Not all the material in the supplement is cited in the main manuscript. Please add text in the manuscript to direct the reader to more information in the supplement.**

A reference to the unreferenced Figures (S5-S8) has been added in Line 519. The remaining unreferenced figure (S2), is referenced directly within the supplement.

**3. Line 367. It should be "exclusively be attributed to" instead of "exclusively to attributed to".**

This typo has been corrected.